# LUBAC modulates CBM complex functions downstream of TRAF6 in T cells

Carina Graß[1,9], Franziska Ober [1,9], Constanze Sixt [1], Bahareh Nemati Moud [1], Irina Antoshkina [1], Frederick Eberstadt[1], Alisa Puhach [2], Göksu Avar [3,4], Antonia Keßler[1], Thomas J. O'Neill [1], Thomas Seeholzer[1], Jan Kranich[5], Thomas Brocker [5], Katja Lammens [6], Michael P. Menden [3,4], Christina E. Zielinski [2,7] & Daniel Krappmann [1,8] ✉

The CARD11-BCL10-MALT1 (CBM) complex drives NF-κB signaling and MALT1 protease activation after T cell receptor (TCR) stimulation, forming a central signaling hub in adaptive immunity. Both linear ubiquitin chain assembly complex (LUBAC), consisting of HOIP, HOIL-1 and SHARPIN, and TRAF6 interact with the CBM complex. Still, the coordinated activity of these E3 ligases in controlling CBM activity remains elusive. Here we demonstrate that LUBAC, unlike TRAF6, is largely dispensable for TCR-induced NF-κB activation in human CD4⁺ T cells. However, HOIP contributes to NF-κB target gene expression and, with TRAF6, modulates MALT1 substrate recognition, influencing T cell responses. Further, LUBAC-mediated conjugation of Met1-linked ubiquitin chains to BCL10 strictly depends on TRAF6, but putative Met1-ubiquitin acceptor lysines in BCL10 serve essential structural roles that limit accessibility within BCL10-MALT1 filaments. Thus, LUBAC acts downstream of TRAF6 to modulate MALT1 substrate recognition and to catalyze BCL10 ubiquitination, which is incompatible with BCL10-MALT1 filament formation.

CD4 T cells initiate an adaptive immunity upon antigen recognition by the cognate T cell receptors (TCR). The CARD11-BCL10-MALT1 (CBM) complex serves as a central signaling hub that channels TCR proximal events to the canonical NF-κB and JNK pathways[1,2]. In addition, CBM complex assembly triggers activation of the MALT1 protease, which modulates immune responses by cleaving substrates involved in cell signaling (e.g. A20, CYLD, HOIL-1), adhesion (Tensin-3), transcription (RelB) and mRNA regulation (e.g. Regnase-1, Roquins, N4BP1)[3]. CARD11 functions as a molecular seed and recruits preformed BCL10-MALT1 complexes, which induces BCL10 oligomerization and filament formation via its CARD (caspase recruitment domain)[4,5]. The MALT1 death

domain (DD) binds to the CARD of BCL10 in a way that the MALT1 C-terminus, containing three immunoglobulin (Ig) domains, the para-caspase (PCASP) domain and two TRAF6 binding motifs (T6BM), protrudes away from the inner BCL10 core filament[6]. BCL10 oligomerization and BCL10-MALT1 binding are both necessary for the recruitment of BCL10 to CARD11, and thus for CBM complex formation after T cell stimulation[6,7]. In cells, assembly of BCL10 into cytoplasmic oligomers coincides with NF-κB activation, indicating the necessity of BCL10 filament formation[4-6,8,9].

TCR stimulation induces binding of the E3 ligases TRAF6 and LUBAC (linear ubiquitin chain assembly complex) to the CBM

---

[1]Signaling and Immunity, Research Unit Signaling and Translation, Molecular Targets and Therapeutics Center, Helmholtz Munich - German Research Center for Environmental Health, Neuherberg, Germany. [2]Department of Infection Immunology, Leibniz Institute for Natural Product Research and Infection Biology, Jena, Germany. [3]Computational Health Center, Helmholtz Munich, Neuherberg, Germany. [4]Department of Biochemistry and Pharmacology, Bio21 Molecular Science and Biotechnology Institute, The University of Melbourne, Parkville, VIC, Australia. [5]Institute for Immunology, Faculty of Medicine, Biomedical Center Munich, Ludwig-Maximilians-Universität, Munich, Germany. [6]Gene Center, Ludwig-Maximilians-Universität, Munich, Germany. [7]Infection Immunology, Friedrich Schiller University, Jena, Germany. [8]Faculty of Biology, Ludwig-Maximilians-Universität, Munich, Germany. [9]These authors contributed equally: Carina Graß, Franziska Ober. ✉e-mail: daniel.krappmann@helmholtz-munich.de

complex, which has been suggested to activate NF-κB signaling[10–13]. Deletion of TRAF6 or destruction of T6BMs in MALT1 abrogates TCR-triggered NF-κB activation[11,14,15]. TRAF6 catalyzes conjugation of K63-linked ubiquitin chains to MALT1, thereby recruiting the TAB-TAK1 kinase complex, which phosphorylates and activates IKKβ to drive canonical NF-κB signaling[16–19]. Further, LUBAC, consisting of HOIP, HOIL-1, and SHARPIN, is recruited to the CBM complex in T cells, and decorates CARD11, BCL10, and MALT1 with M1-linked ubiquitin chains[10,13,20]. While HOIP knock-down (KD) or knock-out (KO) diminishes CBM-dependent NF-κB activation in Jurkat T cells, inconsistent and often contradictory results have been obtained regarding the role of other LUBAC subunits. Whereas one study suggested that HOIP acts independently of its catalytic activity and HOIL-1, but relies on SHARPIN, another report described that the function of HOIP relies on its ligase activity independent of SHARPIN[10,13]. For HOIL-1, a MALT1 substrate, either positive or negative regulatory functions for TCR-induced NF-κB activation have been described[21–23]. TCR-induced NF-κB signaling is mildly reduced in murine thymocytes and CD4+T cells with conditional LUBAC deficiencies. However, LUBAC ablation impaired thymocyte differentiation independent of NF-κB, and the severely reduced viability of peripheral T cells is not connected to CBM functions[24,25].

BCL10 ubiquitination has been implicated in triggering NF-κB signaling as well as BCL10 degradation and concomitant CBM complex destruction[9,13,26–29]. Through mutagenesis, lysines 17, 31 and 63 in the BCL10 CARD were identified as potential acceptor sites for K63- or M1-linked polyubiquitin chains. Mutation of these residues resulted in impaired IKK recruitment and NF-κB activation[13,29]. M1-ubiquitinated BCL10 can serve as a platform for the association to the noncatalytic IKK subunit NEMO/IKKγ via its M1-ubiquitin binding UBAN (ubiquitin binding in ABIN and NEMO) domain[30–32]. Furthermore, conjugation of K48- or K63-linked ubiquitin chains on lysine 31 and 63 can promote removal of BCL10 filaments by selective autophagy[9,29]. Importantly, inducible BCL10 ubiquitination relies on the presence of MALT1 and CARD11, indicating that ubiquitin conjugation depends on CBM complex assembly[13,29].

Here, we directly compare the contributions of LUBAC and TRAF6 to CBM complex assembly and function in primary human CD4 and Jurkat T cells. We reveal that TRAF6 acts upstream of LUBAC and is a prerequisite for linear ubiquitination of BCL10, but LUBAC is largely dispensable for TCR-induced NF-κB signaling. However, TRAF6 and LUBAC control recognition and cleavage of distinct substrates by MALT1. Thus, our functional and structure-guided analyses demonstrate how the coordinated activity of these E3 ligases controls CBM complex assembly and effector functions.

## Results

### TRAF6 but not LUBAC is the main driver of CBM downstream signaling in human CD4 T cells

Transcripts of *TRAF6* and LUBAC subunits *RNF31/HOIP* and *RBCK1/HOIL-1* are expressed in various subsets of human CD4 T cells (Supplementary Fig. 1a). To clarify the necessity of the E3 ligases TRAF6 and LUBAC, we generated knock-out (KO) cells by CRISPR/Cas9 gene editing of primary human CD4+ T cells purified from PBMCs from healthy donors. Using a sgRNA targeting TCRα for optimization of CRISPR/Cas9 KO in CD4+ T cells, we detected loss of TCRα/β surface expression indicative of homozygous TCRα KO in ~50% of CD4+ T cells (Supplementary Fig. 1b). For sgRNA targeting TRAF6, the same protocol achieved TRAF6 downregulation verified by flow cytometry, indicating that TRAF6 was ablated in a subset of CD4+ T cells (Supplementary Fig. 1c). Transfection of sgRNAs targeting TRAF6, HOIP or HOIL-1 resulted in a decline, but not a complete absence of the targeted proteins in pools of primary human CD4+ T cells from different donors (Supplementary Fig. 1c, d). On average, ~60% reduction of TRAF6 protein and ~85%

reduction of HOIP protein was achieved with the respective sgRNAs (Fig. 1a).

We investigated the effects of TRAF6, HOIP or HOIL-1 KOs on NF-κB signaling. In line with previous results, sgRNAs against HOIP or HOIL-1 caused LUBAC instability[33,34] (Fig. 1b). While IκBα degradation and p65 phosphorylation were mildly impaired in sgTRAF6 T cells upon P/I stimulation, NF-κB signaling was largely unaffected after targeting HOIP or HOIL-1. However, IκBα degradation following TNF stimulation was reduced in sgHOIP and sgHOIL-1 cells, indicating effective targeting of LUBAC in CD4+ T cells (Supplementary Fig. 1e). We observed diminished cleavage of HOIL-1 in sgHOIP, and to a lesser extent in sgTRAF6 CD4+ T cells, but otherwise cleavage of MALT1 substrates was not severely altered (Fig. 1b). There was a slight tendency for constitutive cleavage of CYLD and N4BP1 in sgTRAF6 human CD4+ T cells even in the absence of stimulation. The effect was not as pronounced as in mice with conditional TRAF6 deficiency in CD4+ T cells or TRAF6 KO Jurkat T cells[15], likely due to only partial ablation of TRAF6 in the human CD4 T cells.

Bulk analyses by Western blotting may obscure the effects of TRAF6 or LUBAC KO due to residual expression of targeted genes in many cells, and we switched to single-cell analyses for NF-κB signaling. The strong decline of IκBα protein amounts after 30 min P/I stimulation in sgControl-transfected CD4+ T cells was partially abolished after sgTRAF6 transfection (Fig. 1c, Supplementary Fig. 1f). In contrast, sgHOIP or sgHOIL-1 transfection did not perturb IκBα degradation on single-cell level after P/I stimulation in multiple donors but impaired IκBα reduction after TNF stimulation, which was not observed in sgTRAF6-transfected CD4+ T cells (Fig. 1c, d, Supplementary Fig. 1f, g). We also determined p65 phosphorylation on single-cell level after P/I stimulation in human CD4+ T cells (Fig. 1e, Supplementary Fig. 1h). Again, TRAF6 depletion prompted a significant decrease in p-p65 after P/I stimulation in two independent donors, while HOIP and HOIL-1 KO only mildly diminished p65 phosphorylation.

To analyze the impact of TRAF6 and LUBAC on NF-κB translocation in human CD4+ T cells, we performed image stream analyses and determined p65 nuclear accumulation on single-cell level. P/I stimulation led to nuclear accumulation of p65 in CD4+ T cells transfected with sgControl or sgHOIP (Fig. 1f, g). There was a heterogeneous response after sgTRAF6 transfection, which prevented p65 translocation in ~50% of CD4+ T cells, which matches the KO efficiency in the CD4+ T cells (see Supplementary Fig. 1b). Overall, our comparative analyses provide evidence that TRAF6 is the main driver for initiating NF-κB signaling downstream of the CBM complex in human CD4+ T cells.

### TRAF6 and LUBAC control optimal NF-κB gene induction in human CD4 T cells

To determine the effects of TRAF6 and LUBAC on gene induction in human T cells, we performed CRISPR/Cas9 KO of TRAF6 and HOIP in CD4 T cells of four donors for transcriptomic analyses using RNAseq (Supplementary Fig. 2a). The samples cluster according to stimulation condition (untreated (UT) versus CD3/CD28), but no significant clustering with regards to donors or sgRNA transfection was observed (Supplementary Fig. 2b). Almost 300 genes were upregulated after 90 min of TCR/CD28 stimulation ($\log_2$FoldChange $\geq 1.5$ & $p_{adj} \leq 0.05$) (Supplementary Fig. 2c). The most significantly upregulated biological processes in response to TCR/CD28 stimulation were the Hallmark pathway 'TNFA signaling via NFKB' and Gene Ontology (GO) terms associated with T cell and lymphocyte activation (Fig. 2a). 'TNFA signaling via NFKB' was still induced in sgTRAF6 and sgHOIP transfected CD4 T cells, reflecting the partial KO only in a subset of cells (Fig. 2b). However, there was a significant reduction in the induction of 'TNFA signaling via NFKB' pathway in sgTRAF6 but not sgHOIP CD4 T cells. Certain NF-κB-controlled genes only rely on TRAF6 but not HOIP expression (e.g. *DUSP1*, *BTG2*, *MYC*), but induction of other classical

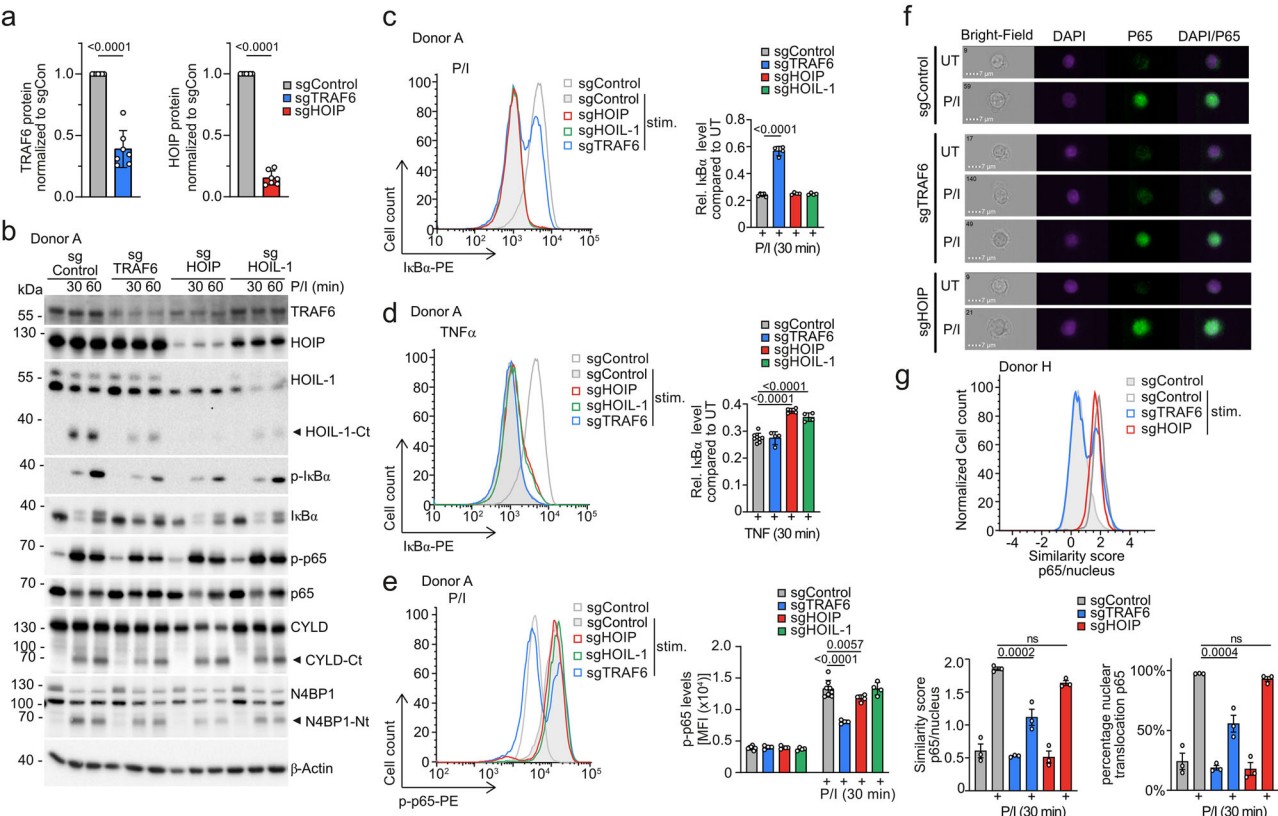

**Fig. 1 | Comparison of TRAF6 and LUBAC-dependent CBM signaling in human CD4+ T cells. a** TRAF6 and HOIP expression normalized to β-Actin in CD4+ T cells from different donors after sgRNA transfection quantified based on Western blotting from 7 biological replicates (see Supplementary Figs. 1c, 2a). **b** Western blot analysis of sgControl, sgTRAF6, sgHOIP and sgHOIL-1 KO primary human CD4+ T cells after stimulation with P/I for the indicated time points. Expression of the target genes (TRAF6, HOIP and HOIL-1), NF-κB signaling and MALT1 substrate cleavage were evaluated by Western blot. **c, d** Representative flow cytometric analyses (left panel) and quantification of changes in mean fluorescence intensity (MFI; values normalized to untreated [UT], right panel) of IκBα levels in sgControl, sgTRAF6, sgHOIP and sgHOIL-1 primary CD4+ T cells after P/I (**c**) or TNF (**d**) stimulation (30 min) from 8 (sgControl) and 4 (targeting sgRNA) biological replicates. IκBα levels of UT samples were set to 1. Treated samples were normalized to the untreated samples. **e** Representative flow cytometric analyses (left panel) and quantification of changes in MFI (right panel) of p-p65 levels in sgControl, sgTRAF6, sgHOIP and sgHOIL-1 primary CD4 T cells after P/I stimulation (30 min) shown for donor A from 8 (sgControl) and 4 (targeting sgRNA) biological replicates. **f** Image stream analyses showing representative pictures. Scale bar: 7 μm. **g** Histograms showing similarity scores of p65 and NucBlue (nucleus) stain and quantification of p65 and NucBlue similarity scores and percentage nuclear p65 of sgControl, sgTRAF6 and sgHOIP transfected CD4+ T cells after P/I stimulation (30 min) from 3 biological replicates. All bars show the means ± SD and p-values were calculated by unpaired two-tailed t-test (**a**) one-way ANOVA (**c, d, g**) or two-way ANOVA (**e**) combined with Dunnett's (**c**–**e**) or Tukey's (**g**) multiple comparisons test. kDa: kilodalton.

NF-κB target genes like *NFKBIA/IκBα*, *TNFAIP3/A20* and *ICAM1* depend on both TRAF6 and HOIP (Fig. 2c, Supplementary Fig. 2d). Of note, TRAF6 but not HOIP KO CD4+ T cells showed mildly augmented 'T cell activation' GO, which may be related to the negative regulatory impact of TRAF6 in naïve CD4+ T cells, and this effect may confine transcriptomic analyses[15,35].

## LUBAC is not a major component of CBM-dependent NF-κB signaling in Jurkat T cells

We have previously shown defective NF-κB activation in TRAF6 KO Jurkat T cells[15]. We decided to study the impact of LUBAC on CBM signaling more closely by generating KOs for each individual LUBAC component in duplicate clones in Jurkat T cells. Again, remaining LUBAC subunits were destabilized in the absence of HOIP, HOIL-1, and to a milder degree SHARPIN (Fig. 3a-c). We transduced an NF-κB-EGFP reporter gene into LUBAC KO clones and measured NF-κB reporter gene activation after P/I, CD3/CD28 or TNF stimulation (Fig. 3d-f, Supplementary Fig. 3a-c). Ablation of HOIP or HOIL-1 nearly abolished, while SHARPIN depletion impaired NF-κB activation by TNF. Accordingly, TNF-triggered IκBα phosphorylation and degradation were decreased in HOIP-, HOIL-1- or SHARPIN-deficient cells, demonstrating the critical necessity of LUBAC for TNFR signaling (Fig. 3g-i). In

contrast, NF-κB reporter induction and IκBα degradation after P/I or CD3/CD28 stimulation was normal or only very mildly reduced in LUBAC KO cells (Fig. 3d-i, Supplementary Fig. 3a-f). Further, neither ERK nor JNK phosphorylation was significantly changed upon P/I or CD3/CD28 treatment after HOIP KO (Supplementary Fig. 3g, h). To confirm these results, we virally reconstituted HOIP and HOIL-1 KO cells (Fig. 3j, l). While reconstitution of FSS-HOIP and FSS-HOIL-1 enhanced TNF-induced NF-κB activation, P/I or CD3/CD28-dependent activation was unaffected or even decreased in the case of CD3/CD28 stimulation after HOIP rescue (Fig. 3k, m). We analyzed *NFKBIA/IκBα* and *TNFAIP3/A20* expression levels in HOIP and TRAF6 KO Jurkat T cells after viral reconstitution and confirmed that HOIP but not TRAF6 is required for induction of both NF-κB target genes after TNF stimulation (Fig. 3n). In contrast, TCR/CD28- and P/I-induced expression of both genes strongly relies on TRAF6 (Fig. 3n, Supplementary Fig. 3i). HOIP did not control immediate induction of NF-κB target genes, but mildly enhanced *TNFAIP3* expression after prolonged TCR/CD28 stimulation.

Since TRAF6 is required for NF-κB signaling in primary CD4+ and Jurkat T cells (Fig. 1)[15], we determined a potential contribution of LUBAC in the absence of TRAF6 and generated HOIP/TRAF6 double (D)KO Jurkat T cells (Fig. 3o). While individual ablation of TRAF6

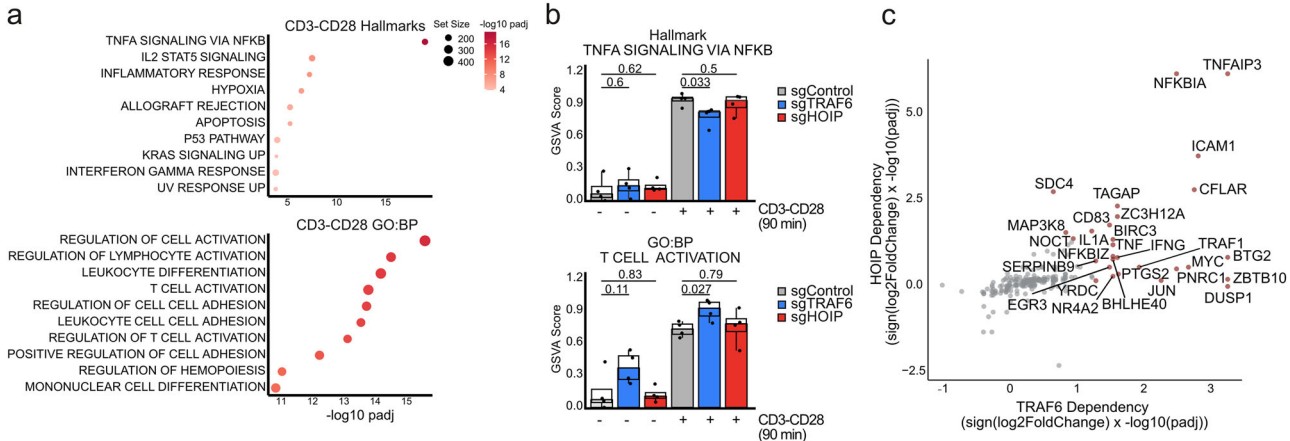

**Fig. 2 | TRAF6 and HOIP-dependent gene expression in CD4 T cells. a** Gene set enrichment analysis (clusterProfiler/fgsea) after CD3/CD28 stimulation, using Hallmark and Gene Ontology Biological Process (GO:BP) gene sets; two-sided enrichment test, upregulated gene sets shown; permutation test p-values, Benjamini–Hochberg adjusted. **b** Gene set variation analysis (GSVA) scores of the Hallmark gene set "TNFA Signaling via NFKB" and GO:BP gene set "T cell Activation". The interquartile range (IQR) around the median was represented, the bars were drawn to the medians, the whiskers extend to 1.5x IQR from the quartiles, and two-sided student's t-test p-values are displayed for the GSVA scores. **c** TRAF6/HOIP dependency analysis ("Methods") using DESeq2 interaction model, two-sided Wald test, Benjamini–Hochberg FDR–adjusted p-values. Higher values indicate weaker induction upon CD3/CD28 stimulation in sgTRAF6 or sgHOIP CD4+ T cells compared to control. All data of RNAseq experiments are from 4 biological replicates (donors).

prevented degradation of IκBα and phosphorylation of p65 after P/I stimulation, the TRAF6/HOIP DKO mirrored the effect of individual TRAF6 deficiency. Taken together, the data suggest that TRAF6 is the main driver for connecting initial CBM activation to the canonical NF-κB signaling pathway after T cell co-stimulation, and LUBAC may contribute to more sustained NF-κB transcriptional responses in Jurkat and human primary CD4+ T cells.

**HOIP and TRAF6 regulate MALT1 substrate selectivity in T cells**

Besides activation of downstream signaling pathways, CBM complex formation leads to induction of MALT1 protease activity, which is constitutively enhanced in the absence of TRAF6[15]. To compare the role of TRAF6 and LUBAC for MALT1 protease activity, we performed biotin pull-downs from Jurkat T cell lysates after incubation of a biotinylated MALT1 activity-based probe (bio-MALT1 ABP) that selectively binds to the active form of MALT1[36]. We quantified the amounts of active MALT1 in unstimulated and P/I treated cells comparing parental, TRAF6, HOIP, TRAF6/HOIP and HOIL-1 KO Jurkat T cells (two clones each) (Fig. 4a). As reported, TRAF6 deletion induced constitutive MALT1 protease activity in Jurkat T cells[15]. HOIP and HOIL-1 ablation did not change constitutive MALT1 protease activity, which was maintained in TRAF6/HOIP DKO cells (Fig. 4a). MALT1 protease was induced to a similar extent in all KO cells after P/I stimulation, despite a mild decrease in the absence of TRAF6.

Next, we tested for MALT1 substrate cleavage after stimulation and noticed that cleavage of HOIL-1 was reduced, while CYLD cleavage was enhanced in HOIP KO cells but unaffected in HOIL-1 KO cells (Supplementary Fig. 4a, b). Therefore, we decided to quantify constitutive and inducible cleavage of MALT1 substrates HOIL-1, CYLD, Regnase-1 and N4BP1 in the panel of Jurkat KO cells (Fig. 4b-d, Supplementary Fig. 4c). Ablation of HOIL-1 did not significantly affect cleavage of any other substrate. Constitutive HOIL-1 cleavage in the absence of TRAF6 as well as P/I-inducible HOIL-1 cleavage, strictly relied upon HOIP expression (Fig. 4b). In contrast, inducible CYLD cleavage was enhanced in the absence of HOIP, which in turn relied on TRAF6 (Fig. 4c). HOIP deficiency did not influence cleavage of Regnase-1 or N4BP1 (Fig. 4d, Supplementary Fig. 4c, d). Further, even though loss of TRAF6 induced constitutive cleavage of all four substrates, inducible HOIL-1 or N4BP1 cleavage was significantly diminished in the absence of TRAF6 (Fig. 4b, d).

To take a closer look by which mechanism HOIP and TRAF6 influence MALT1 substrate selection, we analyzed the cleavage of distinct substrates in KO Jurkat T cells reconstituted with the WT or mutant HOIP or TRAF6 constructs (Supplementary Fig. 4e). While inducible HOIL-1 cleavage was regained by expression of HOIP WT or catalytically inactive C885S, the HOIP-binding mutant ΔUBA failed to rescue cleavage, demonstrating that LUBAC assembly and not E3 ligase activity is required for optimal HOIL-1 cleavage (Fig. 5a). In contrast, augmented CYLD cleavage in the absence of HOIP relied on LUBAC assembly (ΔUBA) and E3 ligase activity (C885S) (Fig. 5b). SPATA2 bridges CYLD to LUBAC[37], but SPATA2 deficiency did not significantly affect IκBα degradation or CYLD cleavage in Jurkat T cells, suggesting that SPATA2 is not involved in the recognition of CYLD by the CBM complex and MALT1 (Supplementary Fig. 4f). In line with the KO data, Regnase-1 cleavage was not influenced by the expression of HOIP WT or mutants (Fig. 5c). In case of TRAF6, expression of TRAF6 WT but neither the dimerization mutant R88A/F118A nor the UBC13-binding mutant C70A rescued the diminished inducible cleavage of HOIL-1, CYLD and N4BP1 (Fig. 5d-f). Since both mutants abolish K63-ubiquitin ligation by TRAF6[38], the catalytic activity of TRAF6 is critical for optimal recognition and cleavage of these substrates.

We also determined the effects of TRAF6 and HOIP deficiencies on A20 cleavage (Fig. 5g). Basal A20 cleavage was enhanced in TRAF6 KO or TRAF6/HOIP DKO cells. Confirming previous results, HOIP facilitates inducible A20 cleavage by MALT1[12]. However, A20 levels are prone to complex regulations, including proteasomal degradation, MALT1 cleavage and TRAF6-NF-κB-dependent resynthesis[39] (Supplementary Fig. 3i). Proteasomal removal of A20 is also impeded in HOIP and TRAF6 KO cells (Fig. 5g), which may indirectly influence the ratio of A20 full length to cleavage fragment. A20 zinc finger (ZnF) 4 and 7 bind to K63- and M1-linked ubiquitin chains, respectively[40]. Thus, we asked if A20 recruitment to ubiquitin chains via the ZnFs controls cleavage of A20 or even HOIL-1 and CYLD by MALT1. We reconstituted A20 KO Jurkat T cells with A20 WT and ZnF mutants (Supplementary Fig. 4g). Cleavage of A20 was severely impaired in ZnF4, ZnF7 or ZnF4/7 mutant expressing Jurkat T cells, suggesting that TRAF6 and HOIP conjugation of K63- and M1-chains, respectively, controls A20 recognition by MALT1 (Fig. 5h). HOIP- and TRAF6-dependent cleavage of HOIL-1 or CYLD cleavage by MALT1 was not affected by A20 ZnF mutations.

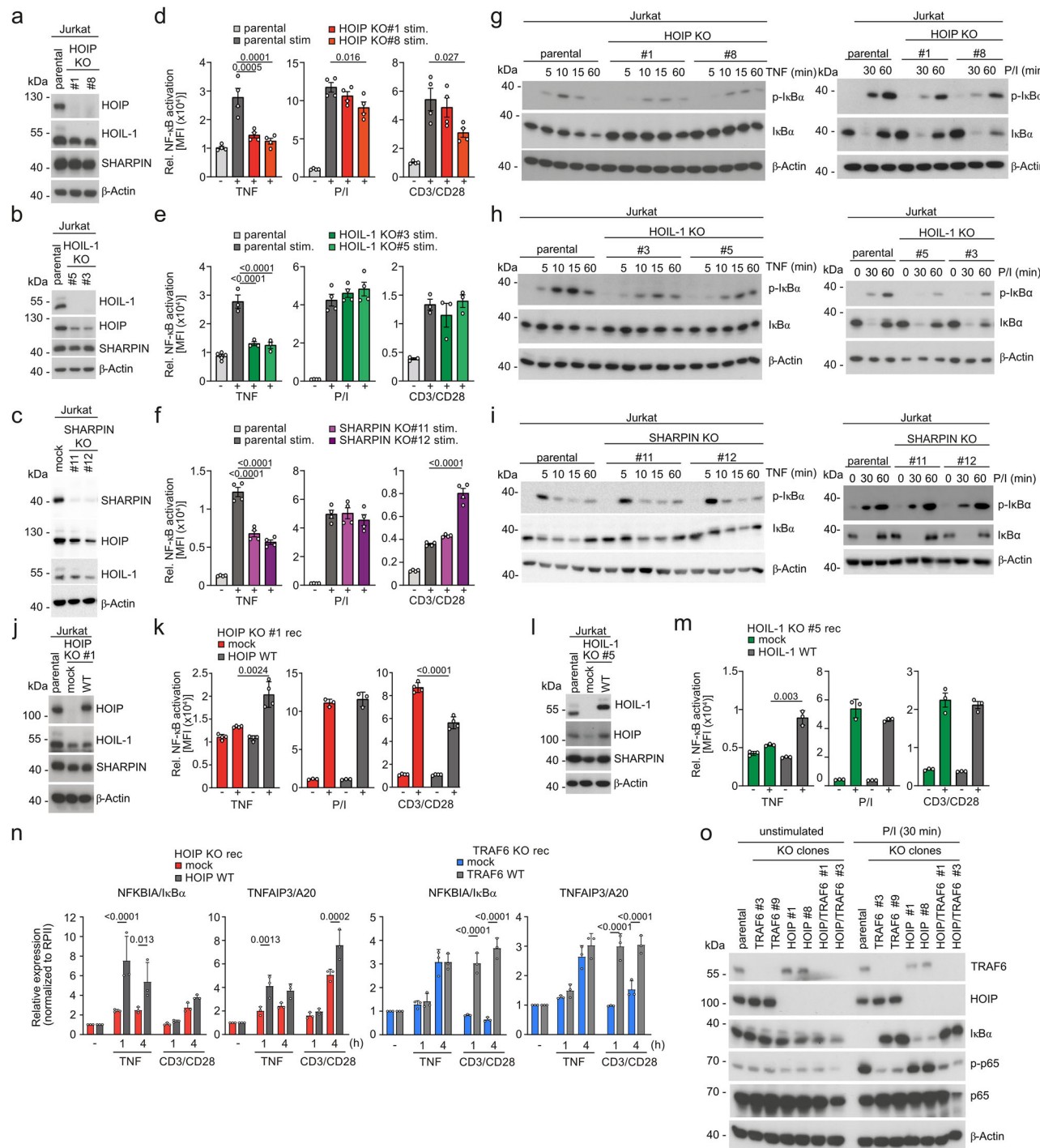

**Fig. 3 | LUBAC is required for TNFR- but not TCR-induced NF-κB activation in Jurkat T cells. a–c** Expression analyses of LUBAC components HOIP, HOIL-1 and SHARPIN in parental and LUBAC KO Jurkat T cell clones by Western blot. **d–f** NF-κB-EGFP reporter activation in parental and LUBAC KO Jurkat T cell clones following stimulation with TNF, P/I and CD3/CD28 stimulation for 5 h. Quantification of flow cytometric EGFP analyses was done by calculating the median fluorescence intensity (MFI) from 4 (d, f, e for P/I) or 3 (e for TNF and CD3/CD28) biological replicates. **g–i** NF-κB signaling was analyzed in parental and KO Jurkat T cells after stimulation with TNF (left panel) and P/I (right panel) for the indicated time points. Western blots showing IκBα phosphorylation and degradation are shown. **j** Reconstitution of HOIP after transduction in HOIP KO Jurkat T cells analyzed by Western blot. **k** NF-κB-EGFP reporter activation in HOIP WT and mock transduced HOIP KO Jurkat T cells following TNF, P/I and CD3/CD28 stimulation for 5 h was analyzed by flow cytometry. Quantification of flow cytometric EGFP analyses was

done by calculating the median fluorescence intensity (MFI) from 4 biological replicates. **l** Reconstitution of HOIL-1 after transduction in HOIL-1 KO Jurkat T cells analyzed by Western blot. **m** NF-κB-EGFP reporter activation and quantification was done as in k from 3 biological replicates. **n** RT-PCR analyses of *NFKBIA/IκBα* and *TNFAIP3/A20* mRNA in HOIP WT and mock transduced HOIP KO (upper panel) and TRAF6 WT and mock transduced TRAF6 KO (lower panel) Jurkat cells following TNF and CD3/CD28 stimulation for 1 and 4 hours from 3 biological replicates. **o** NF-κB signaling was evaluated in parental and TRAF6 KO, HOIP KO and TRAF6/HOIP DKO Jurkat T cells without and with P/I stimulation using Western blot analysis. All bars show the means ± SEM (**d**–**f**, **k**, **m**) or ±SD (**n**) and p-values were calculated by unpaired two-tailed t-test (**k**, **m**), one-way ANOVA combined with Dunnett's multiple comparisons test (**d**–**f**), or two-way ANOVA combined with Sidak's multiple comparisons test (**n**). Only significant p-values (<0.05) are shown. kDa kilodalton.

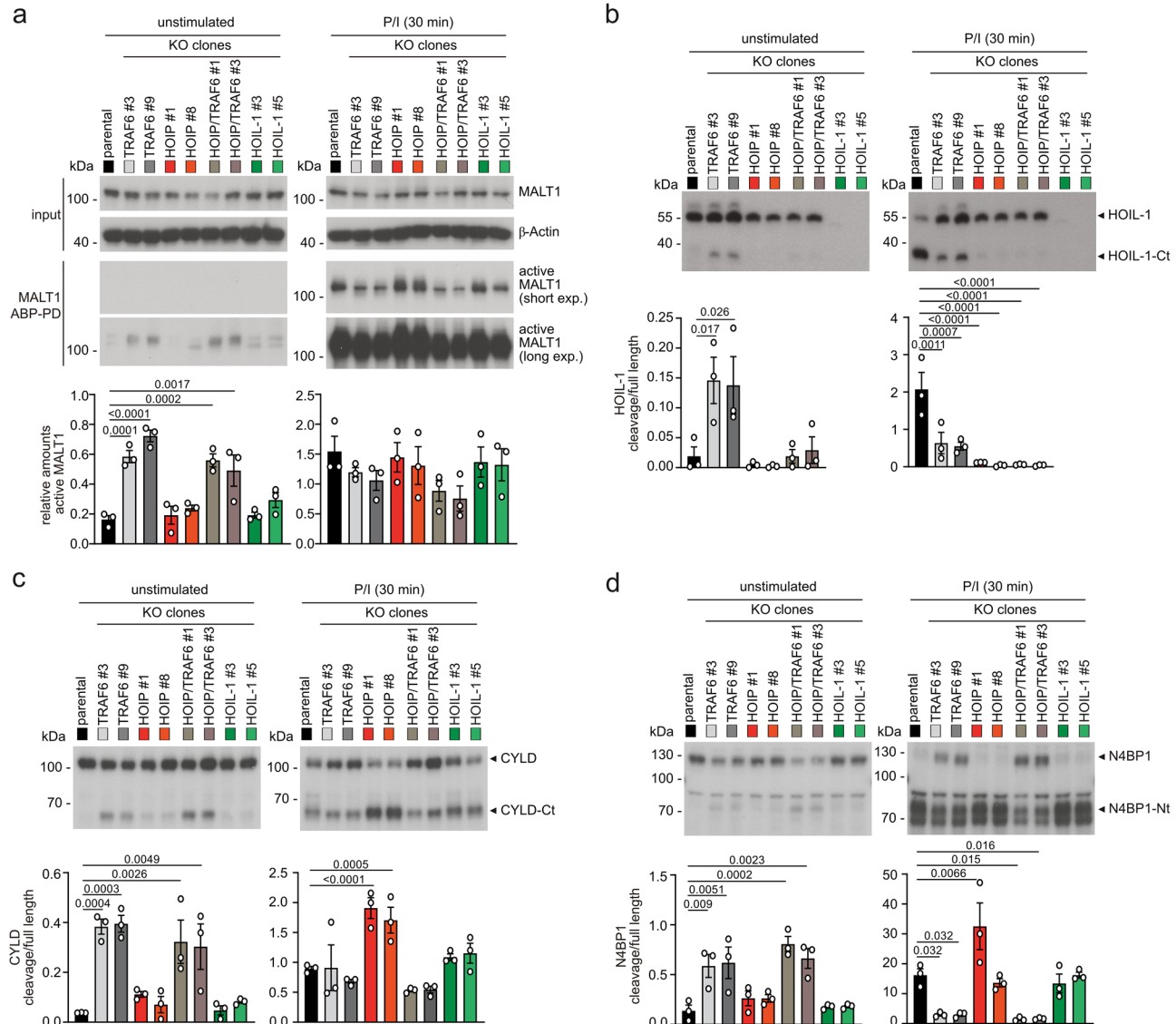

**Fig. 4 | LUBAC and TRAF6 modulate MALT1 substrate recognition. a** Detection of catalytically active MALT1 in parental, TRAF6 KO, HOIP KO, TRAF6/HOIP DKO or HOIL-1 KO Jurkat T cells under untreated or P/I stimulated conditions (30 min) by pulldown of biotin-MALT1-ABP. Substrate cleavage was quantified by determining the ratio of active MALT1 (MALT1 ABP-PD) to total MALT1 (lysate) from 3 biological replicates. **b–d** Cleavage of MALT1 substrates was detected in parental, as well as two independent clones of TRAF6 KO, HOIP KO, TRAF6/HOIP DKO or HOIL-1 KO

Jurkat T cells, either untreated or P/I stimulated (30 min) by Western blot. For quantification the ratios of cleavage products to full-length proteins were determined for HOIL-1 (**b**), CYLD (**c**), and N4BP1 (**d**) from 3 biological replicates. All bars represent the means ± SEM and p-values were calculated by one-way ANOVA combined with Dunnett's multiple comparisons test. Only significant p-values (<0.05) are shown. kDa kilodalton.

Thus, TRAF6 E3 ligase but not LUBAC restricts basal MALT1 protease activity in resting T cells. In addition, TRAF6 catalytic activity enhances cleavage of distinct substrates in activated T cells. LUBAC assembly via HOIP controls cleavage of HOIL-1, but LUBAC assembly and activity also mildly counteract cleavage of CYLD by MALT1.

**N-terminal HOIL-1 cleavage fragment partially rescues LUBAC functions in T cells**
Since HOIL-1 is a substrate of MALT1, CBM signaling may affect LUBAC composition and function. To test the effects in a near to endogenous setting, we virally rescued HOIL-1 KO Jurkat T cells with HOIL-1 1-165 (N-term) and 166-511 (C-term) fragments generated by MALT1 cleavage as well as cleavage resistant (R165A), ligase inactive (C460A) and HOIP-binding defective (ΔUBL) mutants (Fig. 6a). Co-expression of the surface marker ΔCD2 verified successful transduction of all constructs (Supplementary Fig. 5a)[30]. While full length

HOIL-1 proteins were slightly above endogenous HOIL-1 compared to parental Jurkat, expression of HOIL-1 N-term or C-term fragments was lower or higher, respectively. Blockade of protein synthesis by cycloheximide (CHX) did not reveal significant differences in post-translational stability of the fragments, suggesting that other mechanisms counteract strong HOIL-1 N-term expression (Supplementary Fig. 5b). As expected, (see Fig. 3), expression of HOIL-1 WT or mutants did not affect P/I or CD3/CD28-induced NF-κB activation in Jurkat T cells (Fig. 6c). However, HOIL-1 WT, R165A and C460A stabilized HOIP and rescued TNF-triggered NF-κB activation (Fig. 6b, c). HOIL-1 C-term and ΔUBL, which were unable to interact with and failed to stabilize HOIP, did not rescue TNF signaling to NF-κB. In contrast, HOIL-1 N-term, despite low expression, yielded a mild HOIP stabilization and slightly augmented TNF-induced NF-κB response. Thus, the N-terminal HOIL-1 fragment is still able to partially mediate TNFR responses to NF-κB.

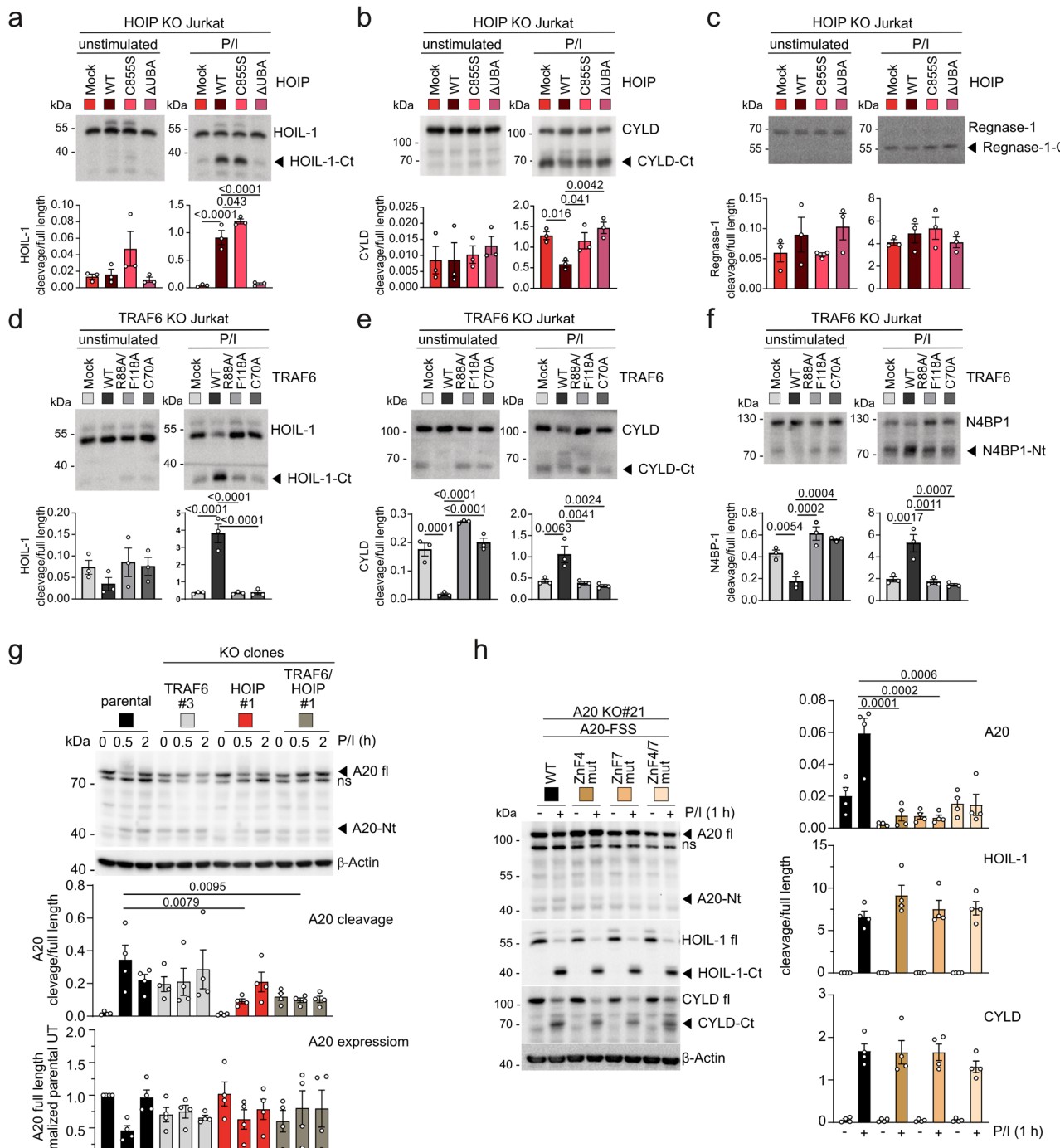

**Fig. 5 | E3 ligase-dependent and -independent modulation of MALT1 substrate recognition by HOIP and TRAF6. a–c** Cleavage of MALT1 substrates HOIL-1 (**a**), CYLD (**b**) and Regnase-1 (**c**) in parental HOIP KO Jurkat T cells transduced with either mock vector, HOIP WT, HOIP C885S or HOIP ΔUBA by Western blot under unstimulated and P/I stimulated (30 min) conditions. Cleavage was quantified as a ratio of cleavage products to full-length protein from 3 biological replicates. **d–f** Cleavage of MALT1 substrates HOIL-1 (**d**), CYLD (**e**) and N4BP1 (**f**) in parental TRAF6 KO Jurkat T cells transduced with either mock vector, TRAF6 WT, TRAF6 R88A/F118A or TRAF6 C70A by Western blot under unstimulated and P/I stimulated (30 min) conditions. Cleavage was quantified as a ratio of cleavage products to full-length protein from 3 biological replicates. **g** Analysis of A20 cleavage in parental, TRAF6 KO, HOIP KO and TRAF6/HOIP DKO Jurkat T cells by Western blot, either

untreated or after stimulation with P/I (30 min and 2 h). The non-specific (ns) band below A20 full length is indicated. A20 full-length amounts were normalized to expression level of β-Actin, and the ratio of cleavage product to full-length was determined from 4 biological replicates. **h** Western blot analyses of A20, HOIL-1 and CYLD cleavage in A20 KO Jurkat T cells reconstituted with A20 WT or ZnF4, ZnF7, ZnF4/7 mutants untreated or after P/I stimulation (1 h). The non-specific (ns) band below A20 full-length is indicated. The ratios of cleavage products to full-length were determined for 4 biological replicates. All bars represent the means ± SEM and p-values were calculated by one-way ANOVA (**a–f**, **h**) or two-way ANOVA (**g**) combined with Dunnett's multiple comparisons test. Only significant p-values (<0.05) are shown. kDa kilodalton.

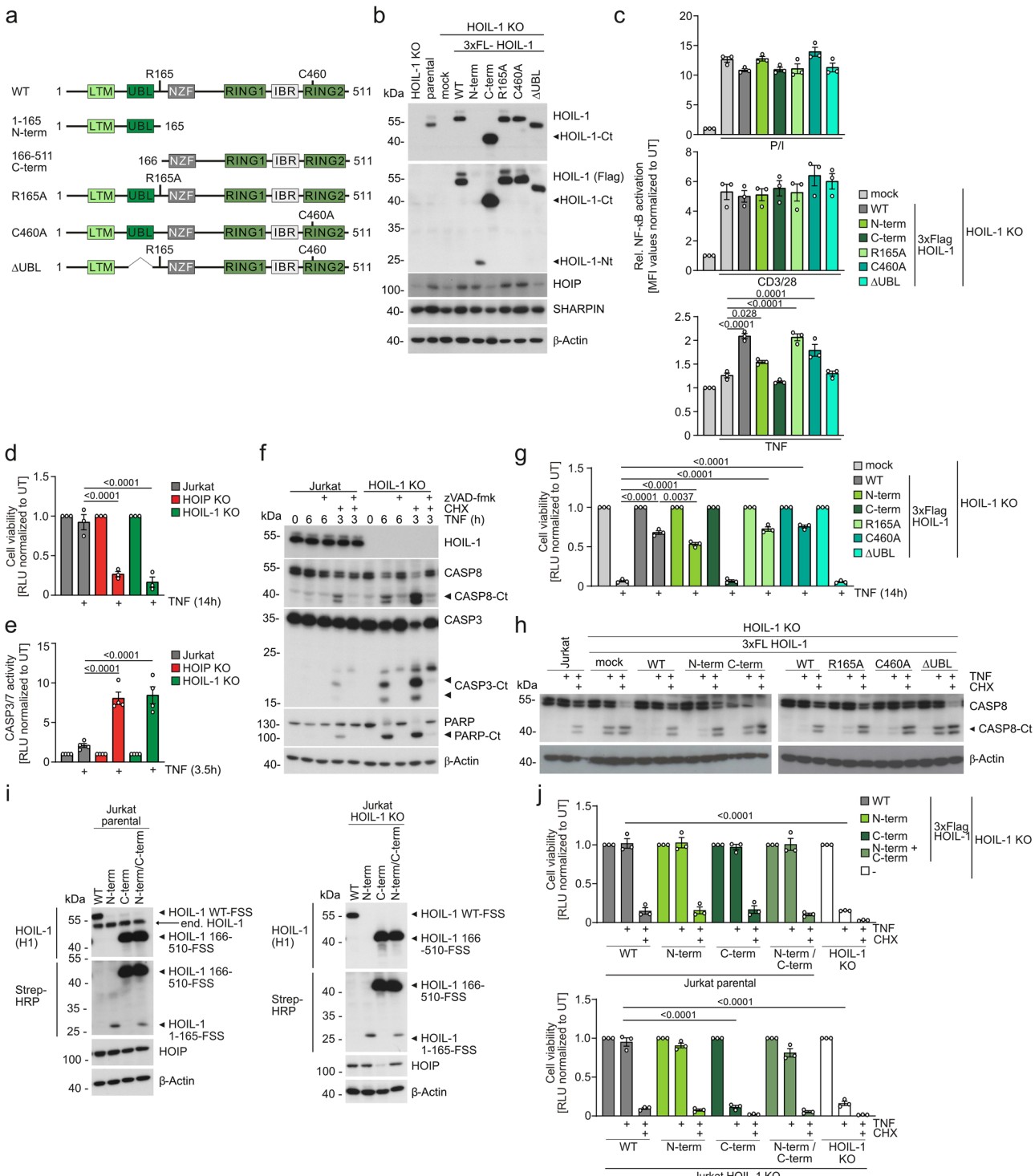

**Fig. 6 | MALT1 cleavage generates a partially active HOIL-1 N-terminal fragment. a** Scheme of HOIL-1 WT and mutant proteins. LTM: LUBAC-tethering motif, UBL: Ubiquitin-like, NZF: Npl4-type zinc-finger, RING: Really interesting new gene, IBR: In-between-RING. **b** LUBAC subunit expression after transduction of HOIL-1 WT and mutants into HOIL-1 KO Jurkat T cells analyzed by Western blot. **c** NF-κB-EGFP reporter induction in HOIL-1 KO Jurkat T cells with HOIL-1 WT and mutant constructs after P/I, CD3/CD28 and TNF stimulation for 5 h. EGFP expression was analyzed by flow cytometry and quantification of median fluorescence intensity (MFI). **d** Cell viability determined by CellTiter-Glo assay after 14 h TNF treatment in parental, HOIP KO and HOIL-1 KO Jurkat T cells. **e** CASP3/7 activation was assessed after 3.5 h TNF treatment in parental, HOIP KO and HOIL-1 KO Jurkat T cells. **f** Parental and HOIL-1 KO Jurkat T cells were treated as indicated. CASP and PARP cleavage was assessed by Western blot. **g** Cell viability determined by CellTiter-Glo assay in HOIL-1 KO Jurkat T cells expressing HOIL-1 WT and mutants following 14 h

TNF stimulation. **h** Induction of CASP-8 cleavage in HOIL-1 KO Jurkat T cells after reconstitution with mock or HOIL-1 WT and mutants after 6 h TNF or TNF/CHX stimulation was analyzed by Western blot. **i** Expression of HOIL-1 WT as well as HOIL-1 N-term and HOIL-1 C-term alone or in combination after transduction into parental (left panel) and HOIL−1 KO (right panel) Jurkat T cells analyzed by Western blot. **j** Cell viability determined by CellTiter-Glo assay in parental (upper panel) or HOIL-1 KO (lower panel) Jurkat T cells transduced with HOIL-1 WT and mutants after TNF or TNF/CHX stimulation for 14 h. All quantifications were done from 3 (**c**, **d**, **g**, **j**) or 4 (**e**) biological replicates after normalization of median fluorescence intensity (MFI; **c**) or relative luminescence units (RLU; **d**, **e**, **g**, **j**) to untreated (UT) samples. All bar graphs represent means ± SEM and p-values were calculated by one-way ANOVA combined with Dunnett's multiple comparisons test. Only significant p-values (<0.05) are shown. kDa kilodalton.

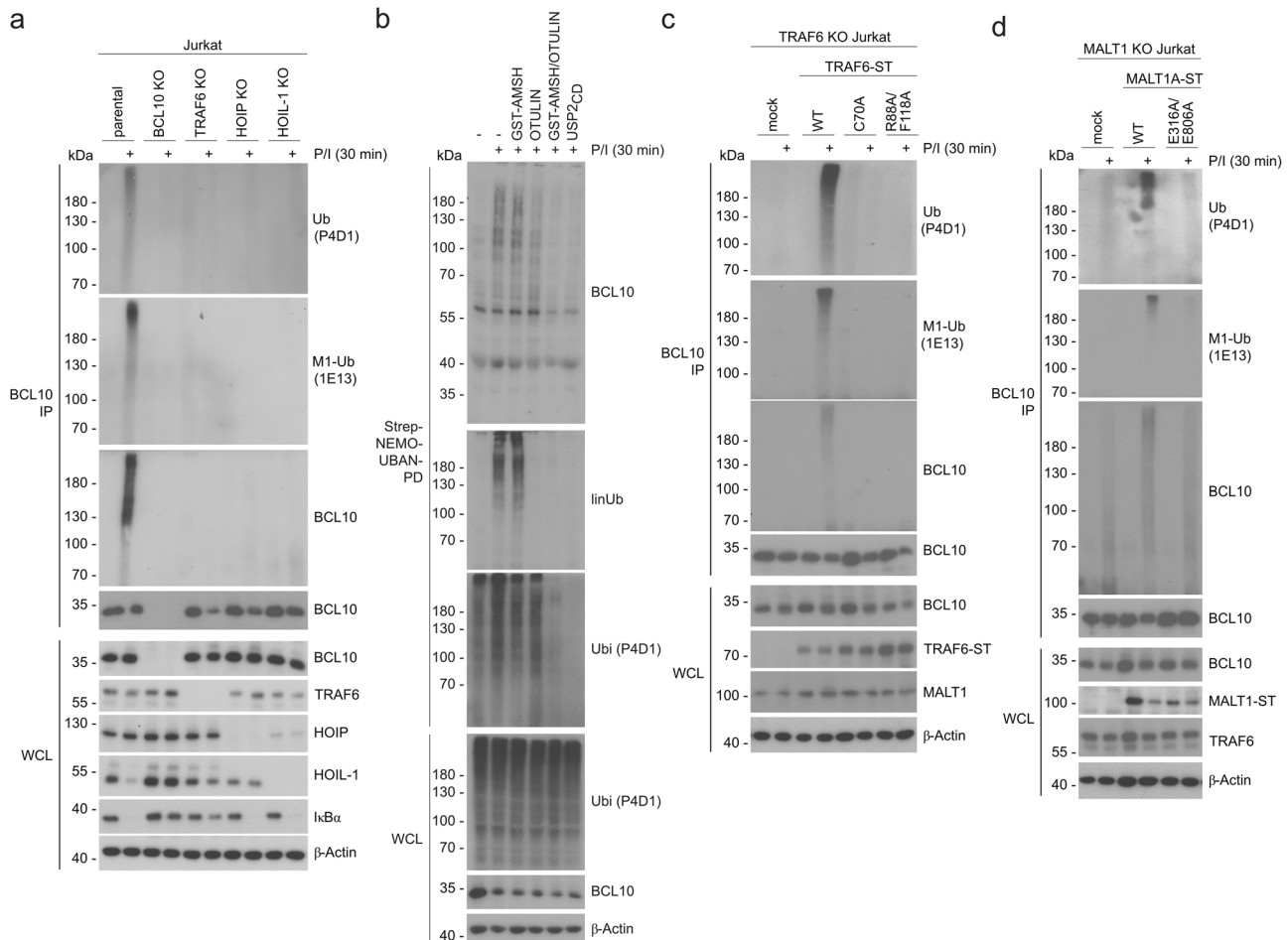

**Fig. 7 | TRAF6-directed K63-ubiquitination enables linear ubiquitination of BCL10 by LUBAC. a** Parental, BCL10 KO, TRAF6 KO, HOIP KO, or HOIL-1 KO Jurkat T cells were stimulated with or without P/I for 30 min. After denaturing lysis, immunoprecipitation (IP) using anti-BCL10 antibody was performed and BCL10 modifications were detected by Western Blot using total ubiquitin (P4D1), M1-ubiquitin (1E13) or BCL10 antibodies. **b** Topology of ubiquitin chains attached to BCL10 was assessed in parental Jurkat T cells after P/I stimulation and pulldown with recombinant Strep-NEMO-UBAN. Immunoprecipitates were incubated with GST-AMSH, OTULIN, GST-AMSH/OTULIN or USP2$_{CD}$, followed by Western blot analysis using the indicated antibodies. **c** TRAF6 KO Jurkat T cells reconstituted with Strep-tagged TRAF6 WT, C70A and R88A/F118A were stimulated with P/I. IP and Western blot analyses were performed as in a using total ubiquitin (P4D1), M1-ubiquitin (1E13) or BCL10 antibodies. **d** MALT1 KO Jurkat T cells reconstituted with Strep-tagged MALT1A WT or E316A/E806A were stimulated with P/I. IP and Western blot analyses were performed as in a using total ubiquitin (P4D1), M1-ubiquitin (1E13) or BCL10 antibodies. Western blots show representative data from two independent experiments. kDa kilodalton.

In non-lymphoid cells, LUBAC is well established to mediate anti-apoptotic NF-κB activation and to counteract the formation of apoptosis and necroptosis-inducing TNFR1 complexes[41]. Ablation of HOIP or HOIL-1 strongly sensitized Jurkat T cells to TNF-induced cell death, which involved activation of Caspases (CASP) 3, 7 and 8 and poly (ADP-Ribose)-Polymerase (PARP) (Fig. 6d-f). Apoptosis inhibitor zVAD-FMK impedes cell death and induces phosphorylation of RIP1, indicating a shift to necroptosis when apoptosis is blocked in LUBAC-deficient Jurkat T cells (Fig. 6f, Supplementary Fig. 5c, d). Next, we tested the ability of HOIL-1 WT and mutants to counteract TNF-induced cell death and CASP8 cleavage in HOIL-1 KO Jurkat T cells (Fig. 6g, h). While HOIL-1 WT, R165A and C460A showed equivalent rescue of cell viability and counteracted CASP8 activation in response to TNF, HOIL-1 C-term and ΔUBL failed to impair cell death. Of note, HOIL-1 N-term was almost as efficient as HOIL-1 WT in counteracting TNF-dependent cell death and CASP8 activation.

Upon TCR stimulation, HOIL-1 is only partially cleaved by MALT1, and all three forms of HOIL-1 (full-length, N-term and C-term) co-exist. Overexpression studies have suggested that the HOIL-1 C-term fragment interferes with LUBAC mediated NF-κB activation[22]. To test a potential impact of the HOIL-1 fragments in trans, we transduced either parental or HOIL-1 KO Jurkat T cells with HOIL-1 WT and the cleavage fragments alone or combined (Fig. 6i). In parental Jurkat T cells expressing endogenous HOIL-1, neither HOIL-1 WT nor HOIL-1 fragments when expressed alone or in combination affected TNF-induced cell death (Fig. 6j). Again, although expressed at very low amounts, HOIL-1 N-term alone or in the presence of HOIL-1 C-term stabilized HOIP (Fig. 6i). Despite high expression, HOIL-1 C-term did not counteract the function of HOIL-1 N-term in impeding TNF-triggered cell death (Fig. 6j). Thus, while the HOIL-1 C-term fragment is inactive, the HOIL-1 N-terminus resulting from MALT1 cleavage is capable of restoring LUBAC functions in counteracting TNFR-induced cell death.

### TRAF6 acts upstream of LUBAC to trigger linear ubiquitination of BCL10

LUBAC recruitment to the TNFR complex relies on cIAP-dependent K63-ubiquitination[42]. Since BCL10 and MALT1 are modified by K63-ubiquitination catalyzed by TRAF6[17,27,29], we asked if TRAF6 may channel linear ubiquitination of BCL10. BCL10 immunoprecipitation under denaturing conditions (1% SDS lysis) detects P/I-induced linear ubiquitination of BCL10 in Jurkat T cells (Fig. 7a). BCL10 ubiquitin chains are pulled-down with the Strep-NEMO UBAN domain, which binds with high

preference to Met1-linked ubiquitin conjugates[30]. Polyubiquitinated BCL10, enriched by Strep-UBAN pull-down, was partially hydrolyzed by Met1-selective DUB OTULIN, but only co-incubation of K63-selective AMSH depleted BCL10 ubiquitin conjugates to the same degree as the pan-ubiquitin hydrolase USP2 (Fig. 7b, Supplementary Fig. 6a). Thus, inducible BCL10 ubiquitination in T cells exhibits a mixed topology consisting of both Met1- and K63-ubiquitin chains.

BCL10 ubiquitination was absent in CARD11 KO and severely reduced in MALT1 KO Jurkat T cells, demonstrating its dependency on CBM complex assembly (Supplementary Fig. 6b). As expected, loss of LUBAC activity in HOIP or HOIL-1 KO Jurkat T cells abolished BCL10 ubiquitination (Fig. 7a). Moreover, TRAF6 deficiency prevented P/I-induced conjugation of M1-linked ubiquitin chains to BCL10, suggesting that TRAF6 acts upstream, controlling LUBAC-catalyzed BCL10 ubiquitination. While reconstitution of TRAF6 KO cells with TRAF6 WT rescued defective BCL10 ubiquitination, TRAF6 catalytically inactive (C70A) and dimerization (R88A/F118A) mutants failed to mediate linear BCL10 ubiquitination (Fig. 7c). To address, whether MALT1 acts as the bridging factor for TRAF6-dependent BCL10 ubiquitination by LUBAC, we reconstituted MALT1 KO cells with MALT1A WT or the MALT1A TRAF6 binding mutant E316A/E806A[11]. Indeed, only MALT1 WT but not MALT1 E316A/E806A rescued the conjugation of linear ubiquitin chains to BCL10 (Fig. 7d). Only shorter ubiquitin chains with non-linear topology were attached to BCL10 in the absence of MALT1 or upon loss of MALT1-TRAF6 interaction. Thus, TRAF6 recruitment to MALT1 and conjugation of K63-linked ubiquitin chains initiates linear ubiquitination of BCL10 catalyzed by LUBAC.

## Ubiquitin acceptor lysines in BCL10 CARD are critical for CBM functions

The CARD of BCL10 contains 13 lysine residues, and in overexpression studies individual or combined mutations K17R, K31R and K63R impaired NF-κB activation and conjugation of M1-, K63- or K48-linked ubiquitin chains[13,29]. We stably transduced BCL10 KO Jurkat T cells expressing an NF-κB-EGFP reporter gene with BCL10 WT or respective ubiquitin-defective mutants and achieved a transduction efficiency >90% (Fig. 8a). BCL10 WT and KK31/63RR proteins were expressed at levels equivalent to endogenous BCL10, but K17R exchange slightly destabilizes BCL10 either as a single mutant or in the KKK17/31/63RRR triple mutant (Fig. 8b). Indeed, ubiquitination of BCL10 was severely blunted by all lysine to arginine exchanges and abrogated by the triple mutant BCL10 KKK17/31/63RRR (Fig. 8c). In line with previous findings, BCL10 K17R or KK31/63RR exchanges severely diminished and the triple mutant abolished the ability of BCL10 to rescue P/I-induced NF-κB activation, but BCL10 was not involved in NF-κB activation after TNF stimulation (Fig. 8d, Supplementary Fig. 7a). Mutation of lysines 17, 31 and 63 in BCL10 abolished induction of NF-κB target genes *NFKBIA* and *TNFAIP3* after TCR/CD28 or P/I stimulation (Fig. 8e). Consistently, IκBα degradation and p65 phosphorylation were impaired in all K/R mutants, with strongest effects in the triple KKK17/31/63RRR BCL10 mutant (Fig. 8f). In addition, cleavage of BCL10 itself, but also other MALT1 substrates such as HOIL-1 and CYLD was reduced or absent in the lysine to arginine exchange mutants (Fig. 8f, Supplementary Fig. 7b). Again, the triple KKK17/31/63RRR mutant most severely decreased NF-κB downstream signaling and MALT1 substrate cleavage.

## Ubiquitin acceptor lysines in BCL10 mediate CBM complex assembly

We investigated the impact of BCL10 R42E or BCL10 L104R mutants, which are known to interfere with BCL10 oligomerization or BCL10-MALT1 dimerization, respectively. Both CBM-destructive mutants impaired NF-κB and MALT1 activation and prevented inducible BCL10 ubiquitination, providing evidence that CBM complex formation is a prerequisite for this modification (Fig. 8c, e, f, Supplementary Fig. 7b)[6,7]. Since extensive networks of interactions are involved in

homotypic BCL10 (CARD-CARD) and heterotypic BCL10-MALT1 (CARD-DD) association in the assembled CBM complex[4–6], we examined the accessibility of lysine residues in context of the BCL10-MALT1 filaments resolved by published cryo-EM (PDB ID 6GK2)[6]. BCL10 filaments form a left-handed helix via homotypic CARD-CARD interactions, and each BCL10 moiety associates with MALT1 via CARD-DD binding in a 1:1 stoichiometry (Fig. 9a). Three interfaces mediate the homotypic CARD-CARD interaction to assemble the helical BCL10 filament structure (Fig. 9b). Interface III is the intra-strand connector that joins two BCL10 moieties within the helix (e.g. connecting BCL10 A1 and B1 subunits). Interfaces I and II are inter-strand connectors between BCL10 subunits in adjacent helical turns (e.g. BCL10 B1-A2 (type I) and BCL10 A1-A2 (type II)). Side chains of the two putative ubiquitin acceptor lysines 31/63 point in the direction of the inner center of either the interface II (K31) or interface I (K63), which form the inter-strand connectors in the BCL10 CARD core filament (Fig. 9b, c). Both K31 and K63 form electrostatic contacts with N75 and E35 in neighboring BCL10 molecules, respectively (Fig. 9c). The localization within the CARD oligomers makes it unlikely that these lysines are accessible for any type of ubiquitination within the tightly packed BCL10 filament. Next, we used molecular modeling to predict the impact of lysine to arginine substitutions at the respective positions (Fig. 9d). Calculations of the energetically preferred positions of arginine rotamers at K31 and K63 predicted a loss of the salt bridges between R63 and E35 as well as R31 and N75, which could affect the oligomeric self-assembly of BCL10. Given the structural constraints in the BCL10-MALT1 filaments and the possible formation of salt bridges to stabilize homotypic CARD-CARD binding, we tested the effects of the KK31/63RR mutations on BCL10-BCL10 binding in HEK293T cells after transient overexpression (Fig. 9e). KK31/63RR mutations in Flag-BCL10 diminished interaction with HA-BCL10 WT to the same degree as BCL10 R42E exchange in α2-helix, which stabilizes the interface I interaction in the BCL10 oligomers by binding to D70 in α4-helix (Fig. 9e)[7].

Lysine 17 is not positioned inside but at the surface of the homotypic BCL10 CARD-CARD interfaces (Fig. 9b). It points away from BCL10 α1-helix in the direction of the α4-helix of the MALT1 DD, which is critical for BCL10-MALT1 interaction (Fig. 9f)[6]. Using overexpression in HEK293T cells, we indeed find that BCL10 K17R exchange alone or in the triple KKK17/31/63RRR mutant reduced the binding of Flag-BCL10 and HA-MALT1 (Fig. 9g). The effect was similar even though not quite as severe as for the destructive BCL10 L104R mutation. To assess the specificity of these lysines, we mutated the stretch of lysine residues K90, K98, K105, K110, K115 (5xK/R) and K118 (6xK/R) in or near BCL10 α-helix 6 involved in BCL10-MALT1 binding (Fig. 9b)[6,43]. In addition, we tested the effects of lysine 118 and 146 single or double mutations, because these residues have been identified as ubiquitin acceptor sites by mass spectrometry in activated B cells[27]. All BCL10 proteins containing single, double or stretches of K-to-R exchanges were expressed at levels of BCL10 WT and the mutations did not perturb conjugation of M1-linked chains to BCL10 or the ability of BCL10 to mediate P/I-induced NF-κB or MALT1 protease activation (Supplementary Fig. 8a-e). Thus, rescue experiments reveal a selective necessity of K17, K31 and K63 in mediating linear ubiquitination of BCL10 as well as NF-κB induction and MALT1 protease activation.

Finally, we determined the effects of lysine to arginine exchanges on inducible CBM complex formation in BCL10 KO Jurkat T cells reconstituted with ubiquitination-defective BCL10 mutants after Strep-PD. BCL10 lysine mutants 5xKR, 6xKR or KK118/146RR, which did not affect BCL10 activity and ubiquitination, did also not exhibit reduced capacity to bind to MALT1 or CARD11 in Jurkat T cells (Supplementary Fig. 8f). However, BCL10 K17R, just like BCL10 L104R, abolished constitutive BCL10-MALT1 binding in T cells, which was intact in BCL10 KK31/63RR (Fig. 9h). Ultimately, all lysine mutations strongly reduced BCL10-MALT1 recruitment to CARD11 and thus

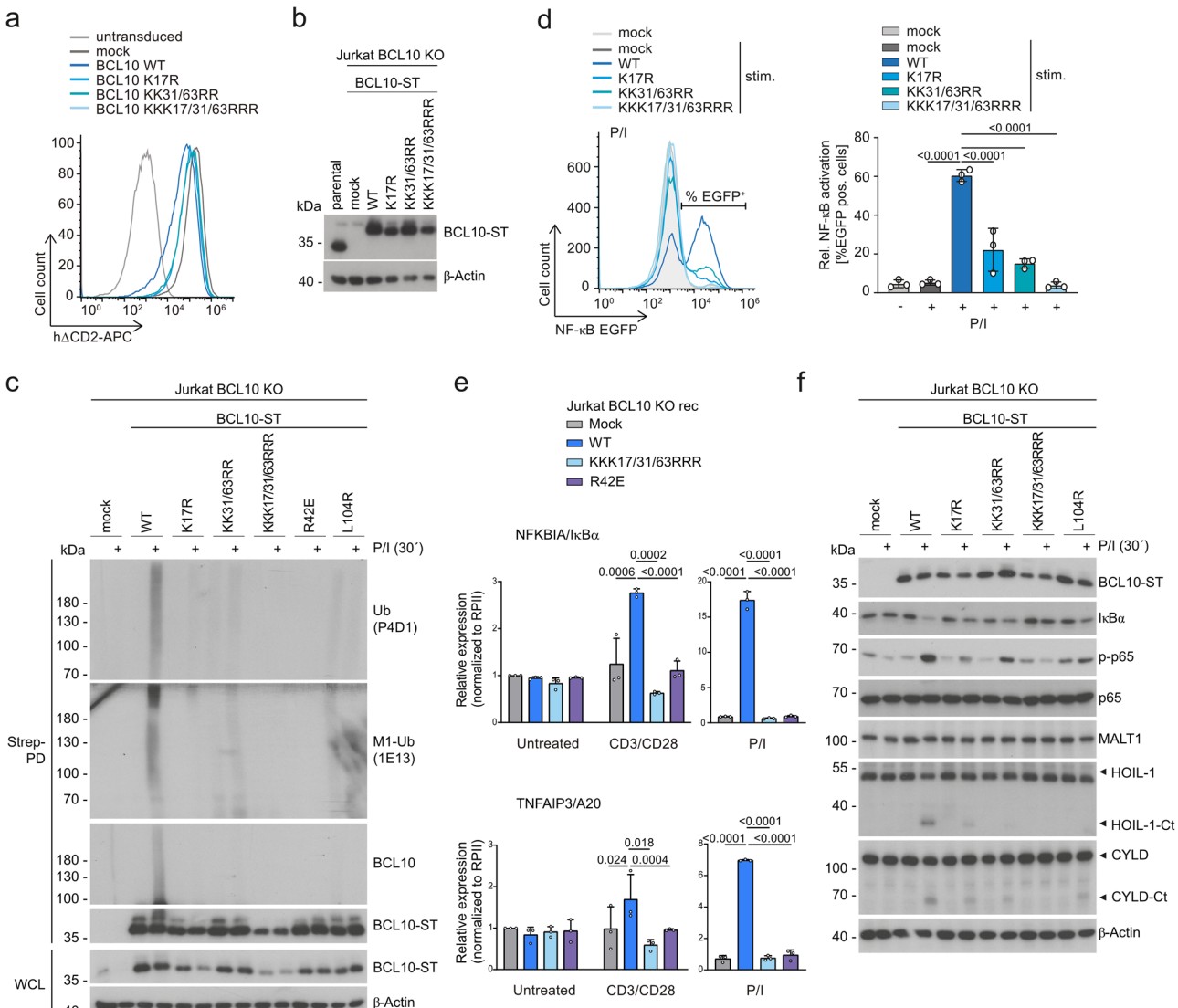

**Fig. 8 | Potential ubiquitin acceptor sites in BCL10 CARD are critical for CBM effector functions. a** Transduction of Flag-Strep-tagged (ST) BCL10 WT and mutants K17R, KK31/63RR and KKK17/31/63RRR in BCL10 KO NF-κB-EGFP reporter Jurkat T cells monitored by the surface-marker hΔCD2 by flow cytometry. **b** Expression of BCL10 WT and mutants was analyzed by Western blot. **c** BCL10 KO Jurkat T cells reconstituted with ST-BCL10 WT and mutants were stimulated with P/I. After denaturing lysis, immunoprecipitation (IP) using Strep-Tactin pulldown (Strep-PD) was performed and BCL10 modifications were analyzed by Western blot using total ubiquitin (P4D1), M1-ubiquitin (1E13) or BCL10 antibodies. **d** NF-κB-EGFP reporter induction of BCL10 KO Jurkat T cells reconstituted with BCL10 WT and mutants after P/I stimulation (5 h) was analyzed by flow cytometry. Representative flow cytometry of EGFP expression is presented in the upper panel and EGFP

expression was quantified determining EGFP positive cells from 3 biological replicates. All bars show the means ± SEM and p-values were calculated by one-way ANOVA combined with Dunnett's multiple comparisons test. **e** *NFKBIA/IκBα* and *TNFAIP3/A20* mRNA expression in BCL10 KO Jurkat T cells reconstituted with BCL10 WT and mutants following 1 h CD3/CD28 or P/I stimulation was determined from 3 biological replicates. All bars show the means ± SD and p-values were calculated by two-way ANOVA combined with Sidak's multiple comparisons test. Only significant p-values (<0.05) are shown. **f** NF-κB signaling (IκBα degradation and p65 phosphorylation) and MALT1 substrate cleavage (HOIL-1 and CYLD) were examined in BCL10 KO Jurkat T cells reconstituted with BCL10 WT and mutants after P/I stimulation by Western blot. kDa kilodalton.

impaired CBM complex formation after T cell stimulation, which explains the complete lack of NF-κB and MALT1 activation. Taken together, lysine residues in the BCL10 CARD may serve as potential ubiquitin acceptor sites, but the same residues are also critical for CBM complex formation.

## Discussion

In this study, we directly compared the effects of LUBAC and TRAF6 deficiency on NF-κB activation in primary human CD4+ and Jurkat T cells. Using single cell analyses we demonstrate that deficiency of LUBAC core components HOIP and HOIL-1 does not significantly influence CBM-dependent NF-κB signaling and translocation after

antigenic stimulation of primary human CD4+ T cells. In contrast, TRAF6 targeting in human T cells impairs NF-κB activation downstream of the CBM complex, which is consistent with defective signaling in TRAF6-deficient murine CD4+ T cells[15,44]. Transcriptomic analyses revealed that TRAF6 but also HOIP contributes to high expression of NF-κB signature genes, suggesting an indirect effect of LUBAC on NF-κB transcription. Conditional inactivation of LUBAC in murine T cells was suggested to mildly impede TCR-induced NF-κB signaling, but it also severely diminished T cell viability[24,25]. Primary human T cells were viable during the analyses, thereby minimizing secondary effects associated with impaired T cell development and viability[15,44]. Further, we generated Jurkat T cell KO clones for each

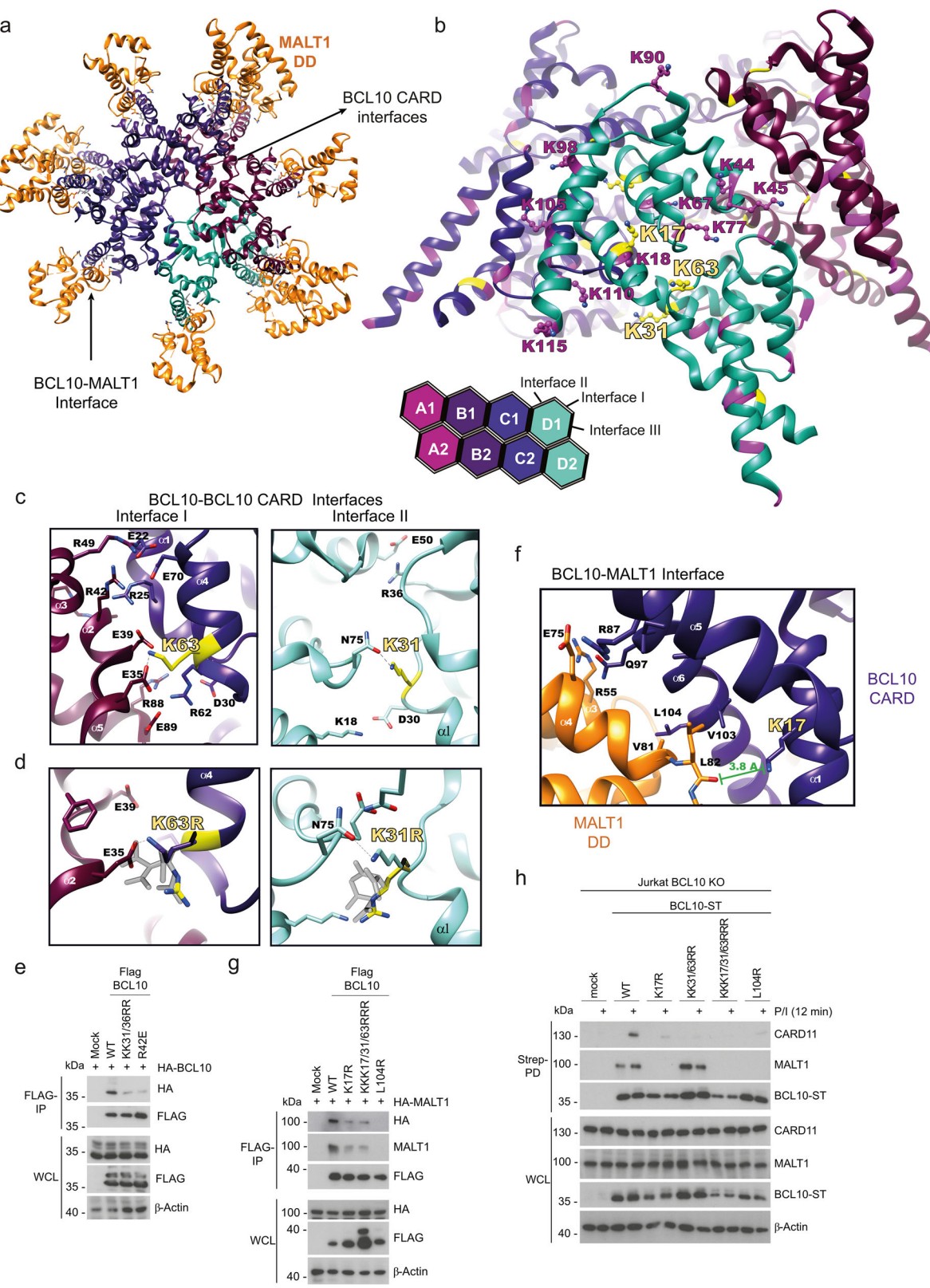

LUBAC subunit and showed that neither TCR-triggered canonical NF-κB signaling nor NF-κB reporter gene induction was impaired in the absence of LUBAC. As previously shown, TRAF6 deficiency nearly abolished NF-κB activation in Jurkat T cells, strictly relying on TRAF6 E3 ligase function[15]. Even combined ablation of TRAF6 and HOIP did not add to the effects of TRAF6 deletion alone. Consistent with previous data on LUBAC deficiency in human and murine fibroblasts[33,45], TNFR-

induced NF-κB activation was reduced but not abolished by HOIP or HOIL-1 ablation in human T cells, suggesting impaired LUBAC functions. Confirming previous results in Jurkat T cells[15], TNFR signaling in primary CD4+ T cells also did not rely on TRAF6.

We can only speculate about the discrepancies to previous studies describing a role of LUBAC for TCR-triggered CBM downstream signaling in Jurkat T cells. Especially, knock-down or knock-out of HOIP

**Fig. 9 | Potential ubiquitin acceptor lysines in BCL10 CARD are critical for CBM complex formation. a** Top view of one helical turn of the BCL10-MALT1 core complex obtained by cryo-EM (PDB ID 6GK2)[6]. MALT1 death domain (DD) is depicted in orange, BCL10 subunits in pink, purple, blue and turquois. BCL10 CARD-CARD interactions and BCL10-MALT1 interface are indicated by arrows. **b** Top: Side view of the BCL10 protein model obtained by the BCL10-MALT1 complex cryo-EM density (PDB ID 6GK2)[6] licensed under CC BY 4.0 (https://creativecommons.org/licenses/by/4.0/CC-BY-4.0). Postulated ubiquitin acceptor sites in BCL10 are highlighted in yellow, other lysine are shown in purple. Bottom: Schematic diagram of the BCL10 helical assembly. Each CARD is represented as a hexagon, and the three interfaces are indicated accordingly. **c** Zoom into BCL10-BCL10 interface I (left) and interface II (right), showing lysines 63 and 31 in yellow, respectively. Interacting residues are shown in stick representation. **d** Molecular modeling of lysine to arginine exchange at K63 (BCL10 interface I) and K31 (BCL10 interface II). Potential positions of arginine rotamers are shown in light grey and the most energetically preferred position in yellow. **e** Flag-tagged BCL10 WT, KK31/63RR and R42E co-expressed with HA-tagged BCL10 WT in HEK293T. After anti-Flag IP, binding of Flag- and HA-BCL10 was analyzed by Western blot. **f** Zoom into the BCL10-MALT1 interface showing K17 of BCL10 juxtaposed to the MALT1 DD. **g** Flag-tagged BCL10 WT, K17R, KKK17/31/63RRR and L104R were co-expressed with HA-tagged MALT1 WT in HEK293T. After anti-Flag IP, binding of Flag-BCL10 and HA-MALT1 was analyzed by Western blot. **h** BCL10 KO Jurkat T cells reconstituted with Strep-tagged (ST) BCL10 WT, K17R, KK31/63RR, KKK17/31/63RRR and L104R were stimulated with P/I. Strep-Tactin pulldown (Strep-PD) was performed and binding of BCL10-MALT1 and BCL10-CARD11 was analyzed by Western blot. Western blots in (**e**, **g**, **h**) show representative data from two independent experiments. kDa kilodalton.

was reported to decrease TCR/CD28-triggered NF-κB activation, which was rescued by reintroducing HOIP[10,13,20]. However, contradictory results have been obtained regarding the requirement of HOIP catalytic activity or the roles of HOIL-1 and SHARPIN in T cells[10,13,20,24,46]. We often find that clonal variations can have a strong impact, especially on T cell co-stimulation, because surface expression of the co-receptor CD28 can be downregulated upon prolonged culturing of cells. PMA/Ionomycin stimulation, which activates the CBM complex by bypassing TCR and co-receptor ligation, was only shown to be decreased in one study using HOIP knock-down[10]. To exclude clonal variations in Jurkat T cells, we stably rescued HOIP and HOIL-1 by viral transduction in KO Jurkat T cells and regained TNFR signaling without augmenting TCR signaling. Thus, the cause of these inconsistencies remains unclear, but our comprehensive analyses in human primary T cells and Jurkat T cell lines provide evidence that TRAF6 is the main E3 ligase that initiates CBM-mediated NF-κB signaling. However, transcriptomic analyses indicate that LUBAC may function as an auxiliary factor for TCR stimulation. In fact, LUBAC dependency may rely on the strength of T cell co-stimulation and modulation of MALT1 substrate cleavage by LUBAC may have long-term effects on T cell activation and differentiation. Our data reveals that BCL10 ubiquitination is not involved in promoting NF-κB activation and the direct mode, how LUBAC affects TCR-dependent T cell activation remains to be determined.

Since LUBAC is recruited to and decorates CBM complex components, especially BCL10, with M1-linked ubiquitin chains, we investigated how LUBAC impacts CBM function[10,12,13,20,27,39]. Even though LUBAC is not crucial to mediate TCR signaling, it modulates CBM functions: (i) HOIP and TRAF6 can either facilitate or hinder cleavage of distinct substrates by MALT1. (ii) LUBAC conjugates ubiquitin chains to the CARD of BCL10, which is incompatible with CBM complex assembly.

(i) With regard to MALT1 protease, we show that HOIP and TRAF6 function in selecting substrates for MALT1 cleavage. HOIP, independent of its ligase activity, is necessary for inducible HOIL-1 cleavage, showing that an intact LUBAC is potentially necessary for HOIL-1 recruitment to MALT1. The positive role of TRAF6 E3 ligase activity for inducible HOIL-1 cleavage is consistent with the observation that TRAF6 acts upstream of LUBAC recruitment to the CBM complex and M1-ubiquitination of BCL10. In contrast, inducible CYLD cleavage is reduced by HOIP and enhanced by TRAF6 activity, showing that conjugation of M1- and K63- ubiquitin chains exercise opposing functions for recognition of CYLD by MALT1. Thus, inactivation of TRAF6 E3 ligase function not only leads to chronic MALT1 protease activation in the absence of acute TCR ligation[15], but K63-conjugation of ubiquitin chains by TRAF6 also impairs stimulation-induced HOIL-1, CYLD and N4BP1 cleavage. Further, proteasome- and MALT1-dependent A20 proteolysis in Jurkat T cells is impeded in the absence of TRAF6 or HOIP, as well as upon mutation of A20 ZnF4 and ZnF7, which bind K63- or M1-ubiquitin chains, respectively. Thus, ubiquitin conjugation and binding are critical for A20 cleavage, suggesting that both E3 ligases

fine tune A20 protein expression at the post-translational level after T cell activation[39]. Cleavage of other substrates like Regnase-1 is unaffected by the absence of TRAF6 or LUBAC. Thus, K63- and M1-ubiquitination by TRAF6 and LUBAC influences MALT1 substrate selection at the CBM complex, revealing a new layer how post-translational modifications can tune recognition and cleavage of individual MALT1 substrates. It will be of interest in how far direct conjugation of K63- or M1-linked ubiquitin chains to MALT1 or BCL10 controls substrate recruitment.

(ii) By ubiquitinating BCL10, LUBAC can modulate CBM complex assembly. BCL10 is a major substrate of LUBAC in activated T cells and lysine-to-arginine exchanges at K17, K31 and K63 in the BCL10 CARD prevent conjugation of M1-linked ubiquitin chains to BCL10[9,13,29]. However, structure-guided and mutational analyses reveal that ubiquitination of these lysine residues is incompatible with the formation of BCL10-MALT1 filaments and CBM complex formation. In fact, unlike LUBAC deficiency, the triple BCL10 KKK17/31/63RRR mutant fails to support MALT1 protease activation, strongly arguing that the cellular effects of these lysine mutations are not connected to loss of linear ubiquitination, but rather destruction of CBM complex assembly. In line with this, BCL10 KK31/63RR fails to form cellular clusters upon TCR stimulation of primary T cells[9]. Since CBM complex formation is a prerequisite for BCL10 ubiquitination[13], it may first seem unlikely that K17, K31 and K63 are direct ubiquitin acceptor sites, but we do not find that surrounding and accessible lysines are involved in linear ubiquitination of BCL10. TRAF6 expression and its recruitment to MALT1 is necessary for BCL10 ubiquitination by LUBAC, suggesting that this modification may serve a secondary function. We can only speculate, but intriguingly, all three putative acceptor lysines are accessible in either the BCL10-MALT1 dimer outside the CBM or at the outer rim of a growing BCL10-MALT1 filament. In fact, BCL10 clusters in activated T cells are co-stained with K63-ubiquitin chains, indicating that the modifications are taking place in the assembling CBM complex[9]. Even though it seems counterintuitive, it may be possible that LUBAC at the assembling CBM complex ubiquitinates 'free' incoming BCL10-MALT1 dimers or modifies BCL10 at the outer end of the growing filament. In both scenarios, LUBAC-catalyzed ubiquitination could block CARD-CARD interactions and prevent new BCL10-MALT1 moieties from oligomerizing, thereby restricting the growth and the length of BCL10 filaments in vivo. Thus, by limiting BCL10 filament growth, LUBAC may counteract CBM signaling. Alternatively, restricting the length of BCL10 filaments may help to concentrate and induce proximity of CBM signaling mediators, such as TAK1 and IKKs, thereby promoting CBM effector functions. Further, the MALT1 protease is activated at the CBM complex, but it can be released and substrate cleavage may also take place at other locations outside the CBM complex and LUBAC could be involved in the dissociation of proteolytic active BCL10-MALT1 dimers[5,36]. All these putatively negative, positive or modulatory functions of LUBAC may not be mutually exclusive, making it difficult to reconcile the impact of LUBAC on CBM complex functions. Further, if

the same residues in BCL10 that are ubiquitinated are controlling BCL10-MALT1 filament assembly, a limitation of the study is that current mutagenesis methods do not allow us to prove that these lysines are the direct ubiquitin acceptor sites. Mass-spectrometry approaches failed to identify ubiquitination at these residues, but this does not exclude that they can be ubiquitinated in activated lymphocytes[27]. However, our combined analyses of BCL10 mutations and LUBAC deficiency provide clear hints that BCL10 ubiquitination can counteract the growth and length of BCL10-MALT1 filaments.

Finally, we demonstrate that LUBAC not only modulates the CBM complex in TCR signaling, but that MALT1-catalyzed cleavage of HOIL-1 alters LUBAC functions in balancing NF-κB and apoptosis signaling after TNF stimulation, opening a new layer of crosstalk between the TCR and TNFR pathways. Since we do not find a major role of HOIL-1 in TCR signaling, we focused on analyzing the functions of HOIL-1 cleavage fragments after TNF stimulation, which is well described to rely on LUBAC activity[47]. The HOIL-1 C-terminal fragment, even when overexpressed, does not act inhibitory on TNFR-induced signaling, despite the fact that C-terminal RBR (RING-between-RING) domain of HOIL-1 can conjugate monoubiquitin and impair LUBAC function[22,48]. Clearly, deletion of the N-terminal UBL domain prevents association and stabilization of HOIP. On the contrary, the short N-terminal HOIL-1 fragment 1-165 generated by MALT1 cleavage contains the UBL, which is sufficient for binding and stabilization of HOIP. Thereby, HOIL-1 N-term partially retains LUBAC functions. Interestingly, while HOIL-1 N-term is only mildly promoting NF-κB activation, it strongly counteracts apoptosis in response to TNF. Thus, MALT1 cleavage is not simply inactivating HOIL-1, but it generates a functional hypomorph that modulates LUBAC functions in balancing NF-κB activation and cell death at the TNFR, revealing a direct crosstalk between the TCR and TNFR complexes. Of note, LUBAC does not antagonize TNF-induced killing in thymocytes, but survival of peripheral T cells relies on LUBAC[24,25]. Our data indicate that upon infection, inflammation or autoimmunity, HOIL-1 cleavage may reduce TNF-induced pro-inflammatory NF-κB activation while maintaining the strong pro-survival function of LUBAC. Lack of phenotypic alterations in mice expressing the cleavage-resistant HOIL-1 R165A mutation indicates that HOIL-1 cleavage may not be relevant under homeostatic conditions[49]. It may be interesting to study if HOIL-1 cleavage contributes to autoimmune inflammation triggered by constitutive MALT1 protease activation or to skin inflammation triggered by CARD14 activating mutations[15,50].

In summary, TRAF6 and not LUBAC is the main E3 ligase driving TCR-induced NF-κB signaling in human T cells. We uncover that HOIP and TRAF6 moderate the recognition of substrates by MALT1 and that linear ubiquitination of BCL10 can counteract BCL10-MALT1 oligomerization, showing that LUBAC tunes CBM complex functions. Moreover, by demonstrating that the N-terminal cleavage fragment of HOIL-1 generated by MALT1 shifts LUBAC functions from NF-κB activation to pro-survival signaling, we reveal a novel mechanism how the TCR can crosstalk to TNF receptors.

## Methods
### Antibodies and DNA constructs
The following antibodies were used for immunoprecipitation (IP) and Western blot (WB): anti-HA (3F1, HMGU core monoclonal antibodies); anti-CARD11 (1D12, #4435, RRID: AB_10694496), anti-p65 (D14E12, #8242S, RRID: AB_10859369), anti-phospho-p65 (93H1, #3033, RRID: AB_331284), anti-IκBα (L35A5, #4814, RRID: AB_390781), anti-p-IκBα (5A5, #9246, RRID: AB_2151442), anti-HOIP (E6M5B, #99633, RRID: AB_2891320), anti-SHARPIN (D4P5B, #12541, RRID: AB-2797949), anti-JNK1/2 (#9252, RRID: AB_2250373), anti-p-JNK (81E11, #4668, RRID: AB_823588), anti-p-ERK (#9101, RRID: AB_331772), anti-Caspase 8 (1C12, #9746, RRID: AB_2275120), anti-Caspase 3 (#9662, RRID: AB_331439), anti-PARP (#9542, RRID: AB_2160739), anti-phospho-RIP1 (Ser166, D1L3S, #65746, RRID: AB_2799693) (all Cell Signaling Technology);

anti-BCL10 (H-197, #sc-5611, RRID: AB_634292), anti-BCL10 (C-17, #sc-9560, RRID: AB_2064858, IP: 2.5 μL), anti-β-Actin (C4, 1:10,000 #sc-47778, RRID: AB_2714189), anti-MALT1 (B-12 for human, #sc-46677, RRID: AB_627909), anti-CYLD (E-10, #sc-74435, RRID: AB_1122022), anti-HOIL-1 (H-1, #sc-393753, RRID: N/A) anti-Ubiquitin (P4D1, #sc-8017, RRID: AB_628423), anti-ERK1/2 (C9, #sc-514302, RRID: AB_2571739) (all Santa Cruz Biotechnology); anti-HOIP (#MAB8039, RRID: AB_10676585), anti-Regnase-1 (#MAB7875, 1:500 RRID: N/A) (R&D Systems); anti-SHARPIN (#14626-1-AP 1:5000, RRID: AB_2187734) (Proteintech); anti-TRAF6 (EP591Y) (#ab33915, RRID: AB_778572), anti-BCL10 (EP606Y) (#ab33905, RRID: AB_725640, IP: 1 μL) (all Abcam); anti-linear ubiquitin (1E3, #ZRB2114, RRID: AB_2938573), anti-FLAG M2 (#F3165, Sigma-Aldrich, WB 1:10,000, IP: 1 μl, RRID: AB_259529) (all Sigma-Aldrich); anti-StrepMAB-HRP (#2-1509-001, RRID: AB_3095590) (IBA Gmbh); horseradish peroxidase (HRP)-conjugated secondary antibodies (anti-rabbit (#711-035-152, RRID: AB_10015282), anti-mouse (#711-035-150, RRID: AB_2340770), Jackson ImmunoResearch); All antibodies used for WB and IP were used at 1:1000 dilution if not otherwise stated.

The following antibodies were used for flow cytometry: anti-hCD2-APC (RPA-2.10, #170029-42, eBioscience, RRID: AB_10805740, 1:400), anti-CD3-FITC (#561806, BD Pharmingen, RRID: AB_11154397, 1:100), anti-CD28-APC (17-0289-41, Invitrogen, RRID: AB_10596352, 1:50), anti-human CD4 FITC antibody (RPA-T4, #300506, BioLegend, RRID: AB_314074, 1:200), IκBα PE-conjugated (L35A5, #7523S, Cell Signaling, RRID: AB_10950821, 1:100), p-p65 PE-conjugated (pS529) (#558423, BD Biosciences, RRID: AB_647222, 1:100), anti-hTRAF6-AF647 (326019, #FAB3284R, R&D Systems, RRID: AB_3649181, 1:200) and anti-hTCRα/β-APC (#306718, Biolegend, RRID: AB_10612569, 1:200). The following antibodies were used for Image Stream: anti-CD16/32 (114-0161-81, eBioscience, RRID: AB_467132, 1:50) and NF-κB p65 XP-Alexa Flour 488-conjugate (D14E12, #49445. Cell Signaling Technology, RRID: AB_2799359, 1:100).

All DNA constructs used are listed in Supplementary Table 1.

### Cell culture and treatments
Human CD4+ T cells were grown in RPMI 1640 medium (#21870076) supplemented with 10% fetal calf serum (FCS; #10270-106), 100 U/ml penicillin/ streptomycin (P/S; #15140122), 1% glutamine (2 mM; #25030081), 1% sodium pyruvate (#11360070), 1% non-essential amino acids (#11140050), and 50 μM β-mercaptoethanol (#31350010) (all Gibco) in the presence of IL-2 (self-made recombinant, 500 U). Parental Jurkat T cells (verified by DSMZ) as well as KO and reconstituted cells, were cultivated in RPMI 1640, Human embryonic kidney (HEK) 293 T cells (DMSZ) in Dulbecco's modified Eagle's medium (DMEM; #41966029) both supplemented with 10% FCS, 100 U/ml P/S (all Gibco) and passaged upon reaching 80% confluency. All cell lines were maintained at 37 °C in a humidified atmosphere supplemented with 5% CO₂. Jurkat T cells were stimulated with 200 ng/ml Phorbol 12-myristate 13-acetate (PMA; Merck Millipore #524400-1) and 300 ng/ml Ionomycin (Calbiochem #407950-1) (P/I) or anti-CD3 (1 μg/ml, #555336) and anti-CD28 (3.3 μg/ml, #555725) together with coupling antibodies anti-mouse IgG1 (1.65 μg/ml, #553440) and anti-mouse IgG2a (1.65 μg/ml, #553387) (all BD Pharmingen) or 40 ng/ml TNF (Sigma #SRP3177). For cell death assays, Jurkat T cells were treated with 100 ng/ml TNF (Sigma) alone or in combination with 2.5 μM Birinapant (Abcam #273619), 50 μg/ml Cycloheximide (CHX, Calbiochem #239764), 10 μM zVAD-FMK (Enzo #ALX-260-020-M001) and 30 μM Necrostatin-1s (Cell Signaling #17802). Primary T cells were stimulated with 200 ng/ml PMA and 300 ng/ml Ionomycin (P/I) or 40 ng/ml TNF if not stated otherwise.

For work with human CD4+ T cells, ethical approval was obtained from the Institutional Review Boards: No. 358/15 and 2025-206-S-SB (Ethikkommission, Technische Universität München), (No. 2020-1985-Material and 2020-2039-Material, Ethik-Kommission,

Universitätsklinikum Jena). All blood donors provided written informed consent. All work was carried out in accordance with the Declaration of Helsinki for experiments involving humans.

## Generation of knock-out Jurkat and primary human CD4+ T cells

Single-guide RNAs (sgRNAs) targeting CARD11, BCL10, TRAF6, HOIP, HOIL-1, SHARPIN, SPATA2 and A20 were cloned into BbsI−linearized px458 vector. Transfection of Jurkat T cells and generation of KO Jurkat clones by serial dilution has been described[51]. sgRNAs targeting SHARPIN were cloned into the lentiCRISPRv2 vector. A control cell line, referred to as mock, was generated using an empty backbone without targeting sgRNA. All sgRNAs used for the generation of KO Jurkat or CD4 T cells are listed in Supplementary Table 2. Lentivirus production and transfection of Jurkat T cells were performed as previously described[52]. 48 h post-transfection, puromycin selection (1 μg/ml, Sigma-Aldrich #540411-25) was initiated. Loss of protein expression of KO cell lines was analyzed by Western blot analysis, CD3/CD28 expression of KO Jurkat T cells was checked via flow cytometry on Attune Acoustic Focusing Cytometer.

For experiments with human CD4 T cells, peripheral blood mononuclear cells (PBMCs) from blood of healthy donors (Donas GmbH, Munich) were freshly isolated by density gradient sedimentation using Ficoll-Paque Plus (GE Healthcare #GE17-1440-02). CD4+ T cells were isolated from PBMCs by negative selection with CD4-specific microbeads (purity ~90%) (Miltenyi Biotec #130-096-533) and were stimulated with plate-bound anti-CD3 (2 μg/ml, clone TR66, Enzo #ALX-804-822-C100) and anti-CD28 (1 μg/ml, CD28.2, BD Biosciences #555725) for 2 days. Purification of CD4+ T cells was verified by flow cytometry (Supplementary Fig. 9a). Cells were stained with anti-human CD4 FITC antibody (1:200 in FACS buffer (3 % FCS in PBS)) and analyzed on Attune NxT Flow Cytometer.

Genes were depleted using TrueGuide Synthetic gRNA (Thermo Fisher Scientific) and Cas9 (HMGU protein production facility) on the third day after isolation. Briefly, 350 ng synthetic gRNA (sgTCRα (#A35534), sgTRAF6 (#A35533), sgHOIP (#A35533) or sgHOIL-1 (#A35533), sgControl (#A35526); see Supplementary Table 3) was mixed with 1.4 μg Cas9 and 7 μl of Resuspension Buffer T (Neon transfection system 10 μL kit) and incubated for 20 min at RT to form a RNP complex. 0.8–1 × 10⁶ activated human CD4 T cells were electroporated and the RNP complexes were delivered into the cells using a Neon transfection system (Thermo Fisher Scientific) with the optimum program (1600 V, 10 ms pulse width, 3 pulses). Electroporated cells were then immediately incubated with pre-warmed RPMI 1640 medium supplemented with FCS, P/S, glutamine, sodium pyruvate, non-essential amino acids, and β-mercaptoethanol in the presence of IL-2. After 5 days of cultivation, expression of proteins was visualized by Western blot to assess the knockout efficiency.

## Flow cytometry analyses of primary human T cells

To assess NF-κB activation following P/I or TNF stimulation, primary human T cells were stained with PE-conjugated IκBα and p-p65 antibodies and analyzed by flow cytometry (Supplementary Fig. 9b). 100,000 cells were seeded and left untreated or treated with TNF or PMA/Ionomycin for 30 min at 37 °C. For IκBα−PE staining, cells were washed with FACS buffer and stained for 30 min at 4 °C with Fixable Viability Dye eFluor 780 (1:1000 in FACS buffer (PBS, 3% FBS), eBioscience #65-0865-18). Cells were fixed in 2% PFA (15 min, RT, Roth #0335.1) and permeabilized in IC buffer (0.1 % saponin (Roth #4185.1) in PBS) (15 min, RT). Fc block (1:50 in IC buffer, Thermo Fisher #14-9161-73) was performed for 7 min at RT. Antibody staining was performed at 4 °C for 30 min in the dark using Iκbα PE-conjugated (1:100 in IC buffer). Cells were washed twice with IC buffer, incubated for 15 min at RT, resuspended in FACS buffer and analyzed on Attune NxT Flow Cytometer. For p-p65 staining, cells were harvested after 30 min

of stimulation and directly fixed in Cytofix Buffer (BD Bioscience #554655) for 10 min at 37 °C. Cells were washed with FACS buffer and permeabilized in Perm Buffer III (BD Bioscience #558050) for 30 min on ice. Afterwards, cells were washed with FACS buffer twice and Fc block was added (1:50 in FACS buffer) for 7 min at RT. Staining was performed at 4 °C for 30 min in the dark using p-p65 PE-conjugated (1:100 in FACS buffer). Cells were washed twice with FACS buffer, incubated for 15 min at RT, resuspended in FACS buffer and analyzed on Attune NxT Flow Cytometer. Data analysis and the generation of histogram overlays were performed using FlowJo software v.10.10 (BD). For IκBα staining, mean fluorescence intensity (MFI) values of the untreated samples in each run were set to 1, and the MFI values of the treated samples were normalized to the untreated samples to obtain relative IκBα levels. For the p-p65 staining, the calculated MFI values of the cells are shown.

## NF-κB p65 translocation assay

To determine the subcellular localization of p65 in primary human T cells, we performed imaging flow cytometry (Image Stream). One million cells per condition were seeded and either stimulated with P/I for 30 min or left untreated. Following stimulation, cells were washed once with cold PBS, incubated in Fixation/Permeabilization Diluent (eBioscience #00-5223-56) for 30 min at 4 °C, washed once with permeabilization buffer (eBioscience #00-8333-56). Fc receptors were blocked with anti-CD16/32 (eBioscience #16-5098-85) in permeabilization buffer, and cells were incubated with an anti-p65 antibody in permeabilization buffer for 30 min at 4 °C. After two washes with FACS buffer, cell nuclei were stained with NucBlue (Thermo Fisher #R37606) for 15 min at room temperature, washed twice with FACS buffer, and finally resuspended in 40 μl FACS buffer for acquisition on an ImageStreamX Mark II (Cytek). Nuclear translocation of p65 was quantified by calculating the overlap between the nuclear and p65 stain, yielding a "similarity score" using the IDEAS software (Cytek). In addition, the percentage of cells with nuclear p65 was determined. Histogram overlays were generated using FlowJo software (BD Biosciences).

## Expression and transcriptomic analyses of human CD4+ T cells

Expression data for TRAF6, RNF31 and RBCK1 is based on Monaco dataset in Protein Atlas (https://www.proteinatlas.org) that performed RNA sequencing (RNAseq) of 29 immune cell types from PBMCs of healthy donors[53]. RNAseq was performed to assess transcriptomic variations of HOIP and TRAF6-depleted primary human CD4 T cells with or without stimulations. 2 Mio cells were seeded and left untreated or treated with CD3/CD28 for 90 min. RNA isolation, quality control, and RNAseq were carried out by Helmholtz Munich Genomics Core Facility. RNA isolation was performed using the Qiagen QiaCube instrument and quantified using Agilent TapeStation. RIN and Qubit-measurement were performed for quality control. RNA sequencing was performed on NovaSeq X Plus (Illumina) in paired-end mode (2×100 bases) with a depth of ≥30 Mio paired reads/sample. 100-bp paired-end reads were aligned to the human genome (GRCh38.p14) using STAR v2.7.11b[54]. Downstream analyses were conducted in the R v4.4.3 environment. Gene-level quantification was performed using Rsubread[55]. Principal component analysis (PCA) and differential expression analysis were carried out using DESeq2[56], with the design formula: ~Donor + Genotype + Treatment + Genotype:Treatment. Multiple hypothesis testing was corrected using the Benjamini-Hochberg[57] method. The Genotype:Treatment interaction term was used to identify genes with differential induction upon CD3/CD28 stimulation in sgTRAF6 or sgHOIP samples compared to sgControl (i.e., Dependency analysis). Genes with $p_{adj} < 0.05$ and negative log₂FoldChange in the corresponding contrasts were classified as TRAF6- or HOIP-dependent. For gene expression visualization, variance-stabilizing transformation (VST) was applied, and donor effects were regressed out using limma[58]. Gene set enrichment analysis

(GSEA)[59] and gene set variation analysis (GSVA) were performed using the ClusterProfiler[60] and GSVA[61] libraries, respectively. GSVA scores were scaled from 0 to 1 for visualization, using min-max normalization. Gene sets were obtained from the MSigDB[62] database via the msigdbr package.

## Lentiviral transduction of Jurkat T cells

For generating stable NF-κB-EGFP reporter cell lines, BCL10 KO, HOIP KO, HOIL-1 KO, and SHARPIN KO Jurkat T cells were lentivirally transduced with pHAGE-Igκ(6x)cona-HygEGFP. Briefly, virus production was conducted by seeding $2.0 \times 10^6$ HEK293T cells in 10 cm² dishes and transfecting them with 1.5 μg psPAX2 (Addgene #12260; gift D. Trono), 1 μg pMD2.G (Addgene #12259; gift D. Trono) and 2 μg pHAGE transfer vector using X-tremeGENE HP DNA Transfection Reagent (Roche #6366236001) on the following day. 72 h hours after transfection, virus-containing supernatant was filtrated (0.22 μM) and added to $5 \times 10^5$ BCL10, HOIP, HOIL-1 or SHARPIN KO Jurkat T cells in the presence of 8 μg/ml polybrene (Sigma-Aldrich #TR-1003-G). 24 h after infection, cells were washed with PBS, resuspended in RPMI and further cultivated. Transduction efficiency was determined by EGFP expression upon TNF, P/I or CD3/CD28 stimulation.

For lentiviral reconstitution of Jurkat T cells, cDNAs were cloned into lentiviral phage vectors coupled to a C-terminal Flag-Strep-Strep (FSS)-tag (BCL10, HOIP, A20, MALT1A) or N-terminal 3xFlag-tag (HOIL-1) (Supplementary Table 1). Human (h)ΔCD2 (lacking the cytoplasmic signaling domain) was co-expressed separated by a co-translational processing site T2A[30]. Expression of hΔCD2 was analyzed by flow cytometry and cells with transduction efficacy >90% were further analyzed. Expression of reconstituted proteins was determined by Western blot. TRAF6 reconstitutions have been described previously[15].

## NF-κB-EGFP reporter assay

To assess NF-κB-EGFP reporter activity, Jurkat T cells were stimulated in 500 μl medium in 24-well-plates with PMA/Ionomycin, anti-CD3/CD28 or TNF for 4-5 h at 37 °C after adjusting the cell number to $0.7 \times 10^6$ cells/ml the day before measurement. Cells were washed and resuspended in 300 μl of PBS and EGFP expression was analyzed by flow cytometry on Attune Acoustic Focusing Cytometer (Supplementary Fig. 9c). Data analysis was performed with FlowJo software v.10.10, either by gating on EGFP-positive cells (Fig. 8d, Supplementary Figs. 7a, 8d) or by calculating the median fluorescence intensity (MFI) of the cells.

## Cell lysis and protein interaction studies

For analysis of expression levels or activation of NF-κB signaling pathways, Jurkat T cells or human primary CD4 T cells ($2-3 \times 10^6$ cells) were lysed in co-immunoprecipitation (co-IP) buffer (25 mM HEPES pH 7.5 (Gibco #15630056), 150 mM NaCl (Sigma Aldrich #106404), 0.2% NP-40 (Sigma Aldrich #74385), 10% glycerol (Roth #4043.3), 1 mM DTT (Roth #6908.1), 10 mM NaF (Sigma Aldrich #S6776), 8 mM β-glycerophosphate (Millipore #35675), 300 μM sodium orthovanadate (Sigma Aldrich #S6508) and protease inhibitor cocktail mix (Roche #11836145001)) for 20 min at 4 °C. After centrifugation, lysates were added to 4x SDS loading dye (Roth #K929.2) and boiled for 5 min at 95 °C. To study endogenous CBM complex formation in Jurkat T cells, $3 \times 10^7$ cells were lysed in co-IP buffer and Strep-tagged BCL10 was precipitated by using 30 μL Strep-Tactin Sepharose beads (1:1 suspension, IBA #2-1201-010) at 4 °C overnight. For BCL10-BCL10 interaction studies, HEK293T cells were transiently transfected with HA-BCL10 and 3xFLAG-BCL10 constructs (pEF vector) using calcium-phosphate precipitation and FLAG-IP was performed by adding anti-Flag-M2 (1 μl) antibody overnight at 4 °C. Lysates were incubated with Protein G Sepharose (15 μl 1:1 suspension, Life Technologies #101242) for 1–2 h at 4 °C to bind respective antibodies. For IPs, beads were washed three times with co-IP buffer followed by the addition of 22 μl 2x SDS loading

dye. Lysates and precipitated proteins were separated by SDS-polyacrylamide gel electrophoresis (SDS-PAGE) and analyzed by Western blot.

## MALT1 activity assay and quantification of substrate cleavage

To assess MALT1 protease activity, $5 \times 10^7$ Jurkat T cells were lysed in 600 μl Co-IP buffer without protease inhibitors. Lysates were incubated with 12 μl high-capacity streptavidin agarose resin beads (Thermo Fisher #20359) for 1 h at 4 °C. After centrifugation ($1700 \times g$, 4 °C) supernatants were transferred and incubated with 0.1 μM biotin-labeled MALT1-activity based probe 7 (MALT1-ABP 7: biotin-KLRSR-AOMK)[36] for 50 min at room temperature. The generation and application of biotin-ABP probe have been previously described[36]. After 50 min rotation, 15 μl high-capacity streptavidin agarose resin beads (Thermo Fisher #20357) were added overnight (4 °C). The following day, beads were washed three times with 500 μl co-IP buffer and denatured in 22 μl 2x SDS loading dye at 95 °C for 8 min.

In addition, MALT1 substrate cleavage of HOIL-1, CYLD, Regnase-1, N4BP1, and A20 was assessed by Western Blot. Quantification of MALT1 activity and substrate cleavage was performed by measuring band intensity on Western blots using Adobe Photoshop or AzureSpot Pro. Values from active MALT1 in the pull-down were divided by total MALT1 in the lysate. A20 full length protein amounts were normalized to expression levels of β-Actin. Furthermore, ratios of the cleaved to full-length substrates were calculated and visualized using Graph Pad Prism.

## BCL10 ubiquitination assays

For detection of ubiquitinated endogenous or Strep-tagged BCL10, $5 \times 10^7$ Jurkat T cells were lysed in 450 μl co-IP buffer (with protease inhibitors) containing 1% SDS at RT. Lysates were homogenized by repeatedly passing through a 20 G syringe and a 26 G syringe, followed by short incubation (20 min, 4 °C, rotating). After 2 centrifugation steps (1: 20,000 x g, 20 min, 4 °C; 2: 45,000 rpm, 20 min, 4 °C) lysates were diluted 1:10 with co-IP buffer without SDS (final SDS concentration: 0.1 %). For BCL10-IP, 1 μl BCL10 antibody (EP606Y) was added overnight (4 °C, rotating) and on the next day 20 μl Protein G Sepharose (50% suspension) for another 1–2 h. For PD of Strep-tagged BCL10, 30 μl Strep-Tactin Sepharose (1:1 suspension) were added overnight (4 °C, rotating). Beads were washed three times with 500 μl co-IP buffer without protease inhibitors (2200 rpm, 3 min, 4 °C), mixed with 25 μl 2x SDS loading dye and boiled for 8 min at 95 °C. Denatured IPs/Strep-PDs and lysates were analyzed by Western blot.

## In Vitro ubiquitin chain linkage analyses

For in vitro ubiquitin chain cleavage assay, 500 ng M1-linked tetra-ubiquitin (R&D Systems #UC-710B-025) or K63-linked Ub4 (R&D Systems #UC-310B-025) were added to 10 μl 1x DUB reaction buffer (500 mM NaCl, 25 mM Tris (pH 7.5), 10 mM DTT) and incubated with 100 nM AMSH-LP (R&D Systems #E-551-050) or OTULIN (R&D Systems #E-558-050) diluted in DUB dilution buffer (150 mM NaCl, 25 mM Tris (pH 7.5), 10 mM DTT) for 60 min at 30 °C. Afterwards, 20 μl 2x SDS loading dye were added and the mixture was boiled for 1 min at 95 °C.

## Protein purification and pull-down

Recombinant Strep-tagged NEMO UBAN (aa257-337) protein[30] was inducibly expressed in BL21(DE3) E. coli cells from the pASK IBA3 vector (IBA Life Science #2-1322-000). The protein was ÄKTA-purified on a 1 ml StrepTrap HP column (GE Healthcare #28-9075-46) followed by desalting on a Superose 6 Increase 10/300 GL column (GE Healthcare #29-0915-96). For Strep-NEMO-UBAN PD, $1 \times 10^8$ Jurkat T cells were stimulated with P/I, and cells were washed with ice-cold PBS and lysed in 800 μL of Ub lysis buffer containing 50 mM Tris (pH 7.5), 150 mM NaCl, 0.5% Triton X-100 (Roth #3051.2), 30 mM N-ethylmaleimide (NEM, Thermo Fisher #23030)), 1 mM DTT, 0.4 mM

sodium orthovanadate, 10 mM NaF, 8 mM β-glycerophosphate, and protease inhibitor mixture for 20 min at 4 °C. Following centrifugation, supernatants were mixed with 20 μg Strep-NEMO-UBAN and rotated overnight at 4 °C. 20 μL Streptavidin beads (Thermo Fisher, #20359; pre-washed 3x with Ub lysis buffer) were added for 2 h (4 °C, rotating). Beads were washed five times with Ub lysis buffer without NEM, all supernatant was removed by vacuum, and 45 μL of Ubicrest reaction buffer was added per sample. 5 μL of 10x DUB (OTULIN, AMSH, OTU-LIN/AMSH or USP2, R&D Systems UbiCREST #K-400) were added and gently mixed, and samples were incubated at 37 °C for 30 min. Samples were centrifuged at 300 x *g*, supernatant was removed by vacuum, and beads were mixed with 25 μL 2x SDS loading dye and boiled for 5 min at 95 °C.

### Western blotting
Proteins separated by SDS-PAGE were transferred onto PVDF-membranes (Merck Millipore #IPVH00010) for immunodetection using an electrophoretic semi-dry transfer system. After transfer, membranes were blocked with 5% milk (Roth #T145.1) in PBS-Tween (PBS-T) (0.01% Tween, Roth #9127.2) for 1 h at room temperature and incubated with specific primary antibodies (dilutions as indicated) in 2.5% BSA (Sigma #A7906) /PBS-T or milk/PBS-T overnight at 4 °C. Membranes were washed in PBS-T before addition of HRP-coupled secondary antibodies (1:7000 in 1.25% BSA/PBS-T or 1.25% milk/PBS-T) for 1 h at room temperature. HRP was detected by enhanced chemiluminescence using the LumiGlo reagent (Cell Signaling Technologies #7003S) according to the manufacturer's instructions and visualized on ECL Amersham Hyperfilms (GE Healthcare #28906839) or recorded using the ECL Chemocam Imager (INTAS) with the ChemoStar Software (INTAS). Images were cropped for presentation using Adobe Photoshop/Affinity Photo. Units for molecular weight markers in kilodalton (kDa) are depicted next to Western blot images.

### RNA isolation, complementary DNA synthesis, and gene expression analyses
$2 \times 10^6$ Jurkat T cells were left untreated or treated with recombinant human TNF (40 ng/ml) or PMA (200 ng/ml) and Ionomycin (300 ng/ml) for the indicated time points at 37 °C. Cells were harvested and RNA was isolated using QIAshredder (QIAGEN #79656) and RNeasy Mini Kit (QIAGEN #5001329) according to the manufacturer's protocol. A total of 1000 ng RNA was reverse transcribed using the Verso cDNA synthesis kit (Life Technologies #AB1453B). For quantitative real-time polymerase chain reaction (qRT-PCR), 25 ng cDNA were mixed with Takyon No ROX SYBR 2x MasterMix blue dTTP (Eurogentec #UF-NSMT-B0710), 1 μM of reverse and 1 μM of forward primer to amplify the genes. Oligonucleotide primers are listed in Supplementary Tab. 4. The annealing temperature for the primers was set to 64 °C and qRT-PCR was performed on a Roche LightCycler 480II instrument and analyzed using Roche LightCycler 480 software 1.5.1.62 SP3.

### Cell viability and caspase activation assays
For the CellTiter-Glo 2.0 viability (Promega #G9242) and the Caspase-Glo 3/7 assay (Promega #G8091), Jurkat T cells were stimulated with 100 ng/ml TNF alone or in combination with CHX, Birinapant (Abcam #273619) or zVAD-FMK (Enzo #ALX-280-020-M001). Cells were seeded in 50 μl medium at a density of 6000 cells per well in 384-well microplates (CulturPlate, PerkinElmer #6007680). After 14 h of treatment, cell viability was assessed by adding 25 μL CellTiter-Glo 2.0 Reagent to each well. Following 10 min of incubation at RT, luminescence signals were measured using the EnVision 2104 Multilabel plate reader (PerkinElmer). Caspase 3/7 activity was measured after 3.5 h of stimulation by adding Caspase-Glo 3/7 Assay Reagent (Promega # G8091) to the wells. After at least 30 min incubation at room temperature, luminescence signals were recorded using the EnVision 2104 Multilabel plate reader (PerkinElmer).

### Protein stability assay
Protein stability of Flag-tagged HOIL-1 WT, HOIL-1 1-165 (N-term) and HOIL-1 166-511 (C-term) was determined by treating Jurkat T cells with CHX (50 μg/ml) to block de novo protein synthesis. Cells were harvested after varying lengths of incubation time with CHX. The stability of HOIL-1 and fragments was assessed by Western blotting with anti-Flag antibody. Protein levels were quantified using LabImage 1D L340 software (INTAS Science Imaging) and normalized to the band of β-Actin.

### Preparation of structural figures
All structural Fig. 9a-f were generated using UCSF Chimera with the cryo EM structure of the BCL10-MALT1 complex filament (PDB entry code 6GK2)[6,63]. The BCL10 interface I and II residues, K63 and K31, and their mutations to arginine were also analyzed with Chimera (Fig. 9). The "swapaa" command was utilized to replace amino acid side chains using data from a rotamer library (Fig. 9d). Rotamers were selected based on the highest probability according to the library and interactively from a rotamer list.

### Statistics and reproducibility
Statistical analyses and data visualization were performed in Graph-PadPrism versions 8.0 and 10.0. All statistical details on statistical tests are provided in the figure legends. Exact p-values are depicted in the figures. Non-specific (ns) p-values ($P \geq 0.05$) are not shown, except wherever it adds important information. Quantification and statistical analyses have been done on at least three biological replicates, except when material was limited (Supplementary Fig. 1f, donor B). Values represent the means ± SEM (standard error of mean) or ± SD (standard deviation) as indicated in the figure legends. Western blots show representative experiments with similar results from at least two biological replicates.

### Reporting summary
Further information on research design is available in the Nature Portfolio Reporting Summary linked to this article.

## Data availability
The raw numbers for charts and graphs are available in the Source Data file whenever possible. Raw reads (FASTQ) of RNA-sequencing data have been deposited in the Sequence Read Archive (SRA) database under BioProject PRJNA1277211. Processed gene-level counts and sample metadata are available in Gene Expression Omnibus (GEO) under GSE299874. All data are included in the Supplementary Information or available from the authors, as are unique reagents used in this Article. The raw numbers for charts and graphs are available in the Source Data file whenever possible. Source data are provided with this paper.

## Code availability
RNA-sequencing analysis code is available on Zenodo (https://doi.org/10.5281/zenodo.17330166) and GitHub (https://github.com/MendenLab/TRAF6_HOIP_huTcells_RNAseq).

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

## Acknowledgements

We thank Kristina Herdt and Katrin Demski for their excellent technical assistance. Arie Geerlof (Helmholtz Munich Protein Expression and Purification Facility) for providing Cas9, Regina Feederle for antibodies (Helmholtz Munich Core Facility Monoclonal Antibodies), the Core Facility Genomics at Helmholtz Zentrum München for technical support. Cas9/sgRNA expression vector PX458 (Addgene #48138) and lenti-CRISPRv2 (Addgene # 52961) was kindly provided by Feng Zhang. psPAX2 (Addgene #12260) and pMD2.G (Addgene #12259) were gifted by Didier Trono. The work was supported by Deutsche Forschungsgemeinschaft (DFG) SFB 1054 (ID 210592381) projects A04 (D.K.), B02 (K.L.), B10 (C.E.Z.) and Z02 (J.K.); SFB 1335 (ID 210592381) projects P07 (D.K.) and P18 (C.E.Z.) and TRR/SFB 124 (ID 210879364) project C7 (C.E.Z.). D.K. received funding from the Deutsche Krebshilfe (project 70115440) and C.E.Z. from the Leibniz Center for Photonics in Infection Research (LPI-BT6) and Carl-Zeiss Stiftung. M.P.M. received funding from the European Union's Horizon 2020 Research and Innovation Program (grant number 950293 COMBAT-RES).

## Author contributions

C.G., F.O. and D.K. designed the study and experiments. C.G. and F.O. performed most experiments, analyzed the data, and wrote the manuscript. C.S. and F.O. designed, performed and analyzed sgRNA targeting in primary T cells, which was supported by B.N.M., A.P., C.E.Z., J.K. and T.B. C.S. performed transcriptomic analyses and G.A. and M.P.M. performed bioinformatic analysis. B.N.M., I.A., F.E., A.K., T.J.O. and T.S. contributed critical experiments. K.L. performed structural analyses and molecular modeling. D.K. supervised the project, analyzed the data and wrote the manuscript. All authors discussed the results and contributed to and reviewed the manuscript.

## Funding

## Competing interests

The authors declare no competing interests.
