## [Transparent Peer Review file · Nature Communications]

LUBAC modulates CBM complex functions downstream of TRAF6 in T cells

Corresponding Author: Professor Daniel Krappmann

Version 0:

Reviewer comments:

Reviewer #1

(Remarks to the Author)

In this manuscript, Graß and colleagues provide additional insights into the role of the Linear Ubiquitin Assembly Complex (LUBAC) within the CARD11-BCL10-MALT1 (CBM) complex of human lymphocytes following antigen receptor stimulation. Using a series of knockout Jurkat cell models and primary human cells, the authors propose that the LUBAC is mostly dispensable for activating NF- κ B, in contrast to the E3 ligase TRAF6. However, the LUBAC catalyzes the attachment of linear ubiquitin chains onto BCL10 and regulates the ability of the paracaspase MALT1 to cleave some of its substrates, i.e., CYLD and HOIL1. Interestingly, the key Lysine acceptors on BCL10 are unlikely to be accessible in BCL10 filaments and appear crucial for the structural organization of the protein. The authors also report that MALT1-mediated HOIL1 cleavage does not affect the activity of the CBM complex. The N-tl fragment resulting from MALT1 enzyme stabilizes the LUBAC and allows TNF α signaling.

- Several independent laboratories have documented the role of the LUBAC during antigen receptor-mediated NF- κ B activation or downstream the CBM complex in thymocytes, B, and T cell lines, using different approaches (knockdown, knockout, knockout-rescue, inhibition) and readouts (biochemistry, qPCR for NF- κ B targets, ...). Nonetheless, the literature data also suggests a regulatory rather than a vital role, especially when compared to the deletion of CARD11, BCL10, or MALT1. Here, although the total degradation of I κ B α doesn't seem to be affected, the presented data still show reduced phosphorylation of I κ B α without HOIP (also seen without HOIL1). In addition, a trend suggesting reduced NF- κ B activation can be seen without HOIP in cells stimulated with anti-CD3 and anti-CD28 antibodies. Previous publications used lower concentrations of stimuli, and a possible explanation might be that the positive role of the LUBAC is rapidly overcome depending on the strength of the stimulus. I agree from the presented data and previous work from the authors that TRAF6 plays a more profound role in TCR-mediated NF- κ B activation. However, I suggest the author consider softening their conclusion on the non-involvement of the LUBAC in NF- κ B activation.

- The description that a hierarchy or a differential selectivity for MALT1 substrates exists is highly interesting as it was thought that all substrates were simultaneously cleaved. To further strengthen this finding, the authors should test additional known substrates such as MALT1, BCL10, RelB, or A20. For instance, it was previously proposed that HOIP knockdown reduced A20 proteolysis by MALT1 (PMID: 24491438). Regnase and N4BP1 are nearly entirely cleaved upon stimulation. It is, therefore, hard to evaluate whether this could be further enhanced without HOIP. Including additional time points would be helpful.

- The abundance of HOIL1 is decreased without HOIP. Could this explain the decreased cleavage by MALT1?

- CYLD connects the LUBAC via SPATA2. Is CYLD binding to the CBM changed without HOIP (and without TRAF6)? Also, what happens in the absence of SPATA2?

- This work nicely confirms the M1-linked ubiquitination of BCL10 and the involvement of the K17, K31, and K63 residues. CARD11 and MALT1 were also shown to undergo M1-linked ubiquitination. Did the authors investigate the potential impact of MALT1 linear ubiquitination on its ability to cleave CYLD and HOIL1?

- The authors may want to further explain the crosstalk between TCR and TNF pathways since MALT1-mediated cleavage

of HOIL1 does not seem to alter TNF-R signaling.

- The data with primary lymphocytes and TNF α (Fig.1B) are hard to interpret. One could also consider there are no overt effects on I κ B α degradation. Moreover, the effect of TRAF6sg, which serves as a positive control, is rather modest.

Reviewer #2

(Remarks to the Author)

Manuscript by Gross and colleagues examines the role of TRAF6 and LUBAC components in Card11-Bcl10-Malt1 (CBM) complex function and signaling. The authors used primary and transformed human cells with knockout of TRAF6 and LUBAC components to explore their importance for CBM induced NF- κ B and MAPK activation and regulation of proteolytic activity of MALT1. The authors conclude that TRAF6 is critical for CBM induced NF- κ B and MAPK signaling while LUBAC play a minor, more supportive role. However, LUBAC is relevant for the modulation of MALT1 proteolytic activity and substrate preference.

Overall, experiments were well designed, technically well executed and conducted with appropriate controls. However, conceptual novelty is minimal as the authors focus on defining and/or confirming details and specifics involving CBM complex signaling such as lysine residues on Bcl10 that are primary targets of ubiquitination, or residues that are important for the proper complex assembly.

What is missing is the biological-functional relevance of signaling events studied in this manuscript. When they knockout TRAF6 and LUBAC components the authors biochemically evaluate and quantify signaling readouts (I κ B α phosphorylation and degradation, or p65 phosphorylation, for example), which is a proper way of evaluating signaling pathways. However, we do not know how this is reflected on NF- κ B mediated gene expression and cytokine production. Therefore, in figures where they biochemically evaluate signaling readouts, the authors should also check gene expression and cytokine production stimulated by CBM or TNFR complexes (this applies to figures 1, 2, 6, S6, etc).

The authors should clearly state that they did not perform cryoEM structural experiments, but they merely interpret or model existing structural data.

Reviewer #3

(Remarks to the Author)

In signaling via lymphocyte antigen receptors, two ubiquitin ligases, TRAF6 and LUBAC, plays crucial roles. In T lymphocytes, ligation of CD3 and CD28 (CD3/CD28) or stimulation with PMA and ionomycin (P/I) induces formation of CARD11-BCL1-10-MALT1 (CBM) complex and elicits activation of NF- κ B and MALT1 protease (paracaspase). The authors dissected the involvement of the two ubiquitin TRAF6 and LUBAC in CBM-mediated signaling in peripheral human T cells and Jurkat T cells. LUBAC is composed of three subunits, HOIP, which is the catalytic subunit for linear ubiquitylation, HOIL-1, and SHARPIN. The catalytic subunit is HOIP and no LUBAC exist in cells lacking HOIP. The amount of HOIP is destabilized in cells lacking HOIL-1 and/or SHARPIN, whereas HOIL-1 or SHARPIN can exist without HOIP. In human cells, although loss of HOIL-1 heavily destabilized HOIP, the small amount of HOIP can exist as HOIP can form complex with SHARPIN. The authors showed that TRAF6 is prerequisite for NF- κ B activation via the CBM complex, whereas LUBAC is dispensable. The authors have also shown that HOIP is involved in the selection of substrates of the MALT1 protease. Some observations described in this manuscript have been reported previously or some molecular mechanism of signaling have already reported in other signaling pathways in which TRAF6 and/or LUBAC are involved. Involvement of both ligases in substrate selection and activation of MALT1 protease is interesting, however, the molecular mechanism underscoring the substrate selection has not been dissected. Thus, the reviewer feels that the results presented in this manuscript might be informative in the field of immunology, however, the manuscript might not be suitable for a high impact general journal such as Nature Communications. The specific comments are listed below.

1. It has been already shown that the linear ubiquitylation activity of LUBAC is dispensable for the CBM mediated NF- κ B activation although TRAF6 is indispensable. The authors showed that using human primary CD4 T cells, LUBAC is dispensable for NF- κ B activation in Fig 1. They tried to knock out the components of LUBAC including HOIP and HOIL-1, however substantial amounts of them still remained. Since LUBAC is shown to function as a scaffold complex to facilitate NF- κ B activation in the CBM signaling pathway despite the dispensability of linear ubiquitylation, the results cannot rule out the possibility that LUBAC plays a role in CBM-mediated NF- κ B signaling as a scaffold. It is worth noting that linear ubiquitylation activity of LUBAC is indispensable for NF- κ B activation in TNF signaling, which is completely different situation from the CBM signaling. The authors tried to evaluate the NF- κ B activation using flowcytometric analysis to detect the amount of I κ B α . However, they did not evaluate the amount of HOIP in sgHOIP treated cells. Therefore, they cannot rule out the possibility that HOIP is still expressing in sgHOIP treated cells because of the inefficient knockout.

2. The authors claimed that loss of HOIP only mildly attenuated NF- κ B activation in Jurkat cells stimulated with P/I or CD3/CD28 in Figure 2D. However, in HOIP KO#8, NF- κ B activation by both P/I and CD3/CD28 is significantly attenuated because of loss of LUBAC and attenuation is more pronounced in CD3/CD28 stimulation, more physiological stimulation than in P/I, which reflects the scaffold function of LUBAC as no LUBAC exist in HOIP null cells. The authors showed that loss of HOIL-1 has virtually no effect on NF- κ B activation by P/I or CD3/CD28. Since in cells lacking HOIL-1, the small amount of HOIP can exist and can form the LUBAC complex with SHARPIN in human cells although the amount of HOIP greatly reduced, which clearly indicated that small amount of LUBAC is enough to activate NF- κ B in

CBM signaling as scaffold. Therefore, the authors' claim that LUBAC is dispensable for CBM-mediated NF-kappaB activation is not supported by the results shown in this manuscript.

3. The observation that loss of HOIP affect selectivity of MALT1 substrates although activation of MALT1 is not affected by loss of HOIP although the authors has not examined the molecular mechanism underlying the selectivity. Among the MALT1 substrates, cleavage of HOIL-1 was heavily impaired in HOIP KO cells. Since LUBAC is recruited to the CBM complex, in which MALT1 exists and is activated, upon stimulation with P/I or CD3/CD28. Thus, it seems likely that HOIL-1 is recruited to the CBM complex and cleaved by MALT1 in the complex. However, in HOIP KO cells, HOIL-1 cannot be recruited to the CBM complex, which may result in the inefficient cleavage of HOIL-1 in HOIP KO cells. Therefore, it is worth examining the molecular mechanism underscoring substrate selection. For example, recruitment of MALT1 substrates are affected by loss of HOIP or TRAF6. Also, it is of interest to know linear ubiquitylation activity of LUBAC is involved in the substrate selection. The reviewer is also very curious to know why loss of TRAF6 induces activation of MALT1.

4. The authors claimed that cleavage of HOIL-1 by MALT1 attenuate LUBAC function. However, it has been already shown previously (Fung et al Front Immunol. 12:749794, 2021 for example).

5. In HOIP KO cells, TRAF6 should be recruited to the CBM complex and conjugate K63 chains. Thus, it is very curious to know why ubiquitylation could not be detected with P4D1 (pan ubiquitin antibody) in HOIP KO cells in Figure 5A.

6. What is delta CD2, at the bottom of page 7?

7. What is MALT1 TBM at the bottom of page 9?

Version 1:

Reviewer comments:

Reviewer #1

(Remarks to the Author)

The authors satisfactorily addressed my concerns.

Reviewer #2

(Remarks to the Author)

The authors have largely addressed reviewers' comments. I would include the data with SPATA2 KO cells in the supplemental.

Reviewer #3

(Remarks to the Author)

The authors have made commendable efforts to address the concerns raised in the initial review. Nevertheless, several critical issues remain unresolved, which merit further clarification and investigation.

1. While the authors attempted to elucidate the impact of HOIP knockout in human CD4⁺ T cells, they did not provide quantitative data on HOIP expression in these cells (Figure 1B). Furthermore, although they assert that HOIP deficiency attenuates TNF- α -mediated NF- κ B activation, the observed reduction in I κ B α levels upon TNF- α stimulation (Figure 1C) appears inconsistent with this claim and warrants further explanation.

2. The authors suggest that HOIP may be dispensable for TCR-mediated NF- κ B activation in primary T cells. However, this notion has already been reported in the literature (Okamura et al., Scientific Reports, 2016), and the current manuscript does not offer substantial novel insights beyond this established finding.

3. The authors have appropriately addressed the substrate specificity of MALT1. The finding that HOIP is required for MALT1 cleavage is intriguing. However, the molecular mechanism underlying this requirement remains unexplored. It is conceivable that A20 is recruited to MALT1 via recognition of M1-linked ubiquitin chains conjugated to the CBM complex, potentially mediated by the ZF7 domain of A20, which is known to bind M1-linked chains. This hypothesis deserves experimental validation.

4. The authors claim that ubiquitin acceptor lysines in BCL10 are essential for CBM complex assembly and function, based on mutational analyses. However, the possibility that these lysine mutations may alter the structural conformation of BCL10, thereby affecting its function independently of ubiquitination, cannot be excluded and should be considered.

Version 2:

Reviewer comments:

Reviewer #3

(Remarks to the Author)

The authors satisfactorily addressed the concerns.

To all reviewers:

We sincerely thank all reviewers for their insightful comments and constructive suggestions. In response, we conducted additional experiments that yielded important findings, allowing us to address and clarify all points raised. Our key results are as follows:

1. **First comparative analysis** — now including transcriptomic data — of how TRAF6 and LUBAC regulate CBM-dependent NF- κ B activation in primary human T cells.
2. **New insights into MALT1 substrate recognition**, revealing that TRAF6 and LUBAC activities govern substrate selection, supported by novel mechanistic data.
3. **A conceptual framework** describing how MALT1-mediated cleavage of HOIL-1 influences the balance between TNFR-induced NF- κ B and pro-survival signaling, highlighting TCR–TNFR pathway crosstalk.
4. **Evidence that LUBAC functions downstream** of TRAF6 recruitment and ligase activity within the CBM complex.
5. **Structure–function analysis** demonstrating that BCL10 CARD ubiquitination and oligomerization are mutually exclusive, indicating that M1-linked ubiquitin chains modulate CBM complex assembly and function rather than promoting signaling directly.

With the addition of new data, we have significantly reinforced our conclusions. We are confident that the revised manuscript now presents the level of novelty and conceptual insight suitable for the broad readership of *Nature Communications*.

We marked the main changes in the text in yellow.

Reviewer #1 (Remarks to the Author):

In this manuscript, Graß and colleagues provide additional insights into the role of the Linear Ubiquitin Assembly Complex (LUBAC) within the CARD11-BCL10-MALT1 (CBM) complex of human lymphocytes following antigen receptor stimulation. Using a series of knockout Jurkat cell models and primary human cells, the authors propose that the LUBAC is mostly dispensable for activating NF- κ B, in contrast to the E3 ligase TRAF6. However, the LUBAC catalyzes the attachment of linear ubiquitin chains onto BCL10 and regulates the ability of the paracaspase MALT1 to cleave some of its substrates, i.e., CYLD and HOIL1. Interestingly, the key Lysine acceptors on BCL10 are unlikely to be accessible in BCL10 filaments and appear crucial for the structural organization of the protein. The authors also report that MALT1-mediated HOIL1 cleavage does not affect the activity of the CBM complex. The N-tl fragment resulting from MALT1 enzyme stabilizes the LUBAC and allows TNFa signaling.

- Several independent laboratories have documented the role of the LUBAC during antigen receptor-mediated NF- κ B activation or downstream the CBM complex in thymocytes, B, and T cell lines, using different approaches (knockdown, knockout, knockout-rescue, inhibition) and readouts (biochemistry, qPCR for NF- κ B targets, ...). Nonetheless, the literature data also suggests a regulatory rather than a vital role, especially when compared to the deletion of CARD11, BCL10, or MALT1. Here, although the total degradation of I κ Ba doesn't seem to be affected, the presented data still show reduced phosphorylation of I κ Ba without HOIP (also seen without HOIL1). In addition, a trend suggesting reduced NF- κ B activation can be seen without HOIP in cells stimulated with anti-CD3 and anti-CD28 antibodies. Previous publications used lower concentrations of stimuli, and a possible explanation might be that the positive role of the LUBAC is rapidly overcome depending on the strength of the stimulus. I agree from the presented data and previous work from the authors that TRAF6 plays a

more profound role in TCR-mediated NF- κ B activation. However, I suggest the author consider softening their conclusion on the non-involvement of the LUBAC in NF- κ B activation.

We acknowledge that other laboratories came to other conclusions. We therefore tried to look at the role of LUBAC from different angles such as KO in primary human T cells, KO and reconstitution in Jurkat T cells, reconstitution of potential M1-attachment sites in BCL10. There are some differences in p-I κ B α levels upon KO of LUBAC components, but this readout is sometimes misleading, because it relies on the amounts of cellular I κ B α , which is prone to rapid degradation after stimulation.

To more rigorously test the role of TRAF6 and LUBAC in human CD4 T cells, we performed **single cell analyses of p65 translocation**, which confirmed a strong impact of TRAF6, while HOIP seems largely dispensable (**new Figure 1E, F**). However, we also included **transcriptomic profiling** in human TRAF6 and HOIP KO CD4 T cells after 90 min CD3/CD28 stimulation, indicating an influence of LUBAC on induction of some NF- κ B target genes (**new Figure 1G-I, S2**).

We **revised the Abstract and Discussion** (page 14: 402-409; page 15: 436-441), to take account of the effects of HOIP KO on NF- κ B target genes. We emphasize an auxiliary role of LUBAC, which may depend on the strength of stimulation and also involves modulation of CBM assembly (see Figure 7) and MALT1 substrate recognitions (see Figure 3).

- The description that a hierarchy or a differential selectivity for MALT1 substrates exists is highly interesting as it was thought that all substrates were simultaneously cleaved. To further strengthen this finding, the authors should test additional known substrates such as MALT1, BCL10, RelB, or A20. For instance, it was previously proposed that HOIP knockdown reduced A20 proteolysis by MALT1 (PMID: 24491438). Regnase and N4BP1 are nearly entirely cleaved upon stimulation. It is, therefore, hard to evaluate whether this could be further enhanced without HOIP. Including additional time points would be helpful.

We thank the reviewer for pointing out that distinct MALT1 substrate selectivity is a highly intriguing concept. Indeed, differential recruitment of substrates to the CBM complex would help to explain how substrates are selected, since the recognition site itself seems not to be sufficient. We have not been able to obtain Western Blots sufficient for quantification of all substrates. Especially (auto-) cleavage of MALT1 and the C-terminal BCL10 are difficult to detect and quantify. Detection of the RelB cleavage fragment relies on proteasomal inhibition, which makes it difficult to obtain convincing data on whether TRAF6 or LUBAC control the efficacy of cleavage.

However, we **now quantified A20 cleavage (new Figure S4E)**. We confirm that inducible A20 cleavage is HOIP dependent and TRAF6 depletion leads to constitutive A20 cleavage. As a caveat, A20 is also prone to proteasomal degradation and NF- κ B-dependent resynthesis. Proteasomal degradation is also impeded in HOIP KO cells and mRNA induction relies on TRAF6/NF- κ B signaling (Yin et al., 2022, 35099607; **Figures S4E and S3I**). We monitored cleavage at two time points (30 min and 2 h), but the proteasomal degradation of A20 in parental Jurkat may lead to overestimate efficacy of A20 cleavage, making it difficult to judge the direct influence of HOIP and TRAF6. Nevertheless, we included new data showing that proteasome- and MALT1-dependent A20 proteolysis is controlled by TRAF6 and HOIP.

We also **monitored Regnase-1 and N4BP1 cleavage 0.5 and 2 h** after stimulation and could not see significant changes in HOIP KO Jurkat T cells (**new Figure S4D**). Further, **reconstitution of HOIP KO cells** confirms that Regnase-1 is cleaved independently of LUBAC assembly or E3 ligase activity (**new Figure 3G**). We already measured substrate cleavage at a rather early time point (30 min), to monitor initial recognition and exclude secondary effects and we have not been able obtain convincing differential quantifiable cleavage data at even earlier time points.

- The abundance of HOIL1 is decreased without HOIP. Could this explain the decreased cleavage by MALT1?

We quantified the ratio between cleaved and uncleaved substrates, to account for differences in substrate expression, which is especially apparent for HOIL-1 in HOIP KO cells. The data show that HOIL-1 is not only expressed lower, but also inducible cleavage is nearly abolished by HOIP deficiency and severely reduced in the absence of TRAF6.

Importantly, the questions inspired **new experiments on the mechanism(s), how HOIP and TRAF6 control substrate cleavage** by reconstituting KO cells with WT and mutant constructs (**new Figures 3E-J**).

Indeed, HOIL-1 cleavage by MALT1 relies on LUBAC assembly, but not HOIP E3 ligase activity (**Figure 3E**). Thus, HOIP controls HOIL-1 cleavage by mediating the recruitment to the CBM complex. In contrast, impaired CYLD cleavage relies on HOIP E3 ligase activity (**Figure 3F**), showing that HOIP uses different mechanisms to control HOIL-1 and CYLD cleavage. Again, Regnase-1 cleavage is not influenced by HOIP WT or mutants (**Figure 3G**).

In the case of TRAF6, the efficacy of inducible HOIL-1, CYLD and N4BP1 cleavage relies on UBC13 recruitment (C70A mutant) and dimerization (R88A/F118A mutant), which both mutations lead to defective TRAF6 E3 ligase activity (**Figure 3H-J**). In case of HOIL-1, this is highly consistent with the observation that M1-ubiquitination of BCL10 and thus LUBAC recruitment to the CBM complex depends on TRAF6 E3 ligase activity (**Figure 5D**) and apparently conjugation of K63 chains also recruits other substrates to MALT1.

Overall, the new data on the mechanism of MALT1 cleavage add important insights into how HOIP and TRAF6 control MALT1 substrate cleavage. We have **amended the Results** (page 8/9: 218-241) and **Discussion** (page 16: 448-466) accordingly.

- CYLD connects the LUBAC via SPATA2. Is CYLD binding to the CBM changed without HOIP (and without TRAF6)? Also, what happens in the absence of SPATA2?

We have been trying to use IP to monitor HOIP- and TRAF6-dependent recruitment of endogenous CYLD and other substrates to MALT1, but we have not been able to reliably detect these most likely very transient interactions.

We **generated SPATA2 KO Jurkat T cells**, and we do not see that CYLD cleavage is significantly influenced, suggesting that SPATA2 is not involved in the recognition of CYLD by the CBM complex and MALT1. I κ B α phosphorylation and degradation were also not affected by SPATA2 KO. We include the

data here as additional information for the reviewers.

However, we feel that an on the role of SPATA2 and CYLD goes beyond the scope of this manuscript.

Additional figure for reviewers: Three clones of SPATA KO Jurkat T cells were stimulated with P/I and I κ B α phosphorylation and degradation as well as CYLD cleavage were analyzed by Western blotting.

- This work nicely confirms the M1-linked ubiquitination of BCL10 and the involvement of the K17, K31, and K63 residues. CARD11 and MALT1 were also shown to undergo M1-linked ubiquitination.

Did the authors investigate the potential impact of MALT1 linear ubiquitination on its ability to cleave CYLD and HOIL1?

MALT1 linear ubiquitination has been described in Oikawa et al 2020 (33329596), but the conjugation sites have not been mapped. However, it is **not known to which sites linear ubiquitin chains are attached in MALT1**. With one exception (30741923), mass spectrometry performed by us (unpublished) and by other laboratories so far failed to identify critical ubiquitin acceptor lysines in MALT1. Thus, currently we cannot use mutagenesis to address this question. Nevertheless, it is an intriguing idea. We have therefore **adjusted and extended Discussion** on page 16: 448-466.

- The authors may want to further explain the crosstalk between TCR and TNF pathways since MALT1-mediated cleavage of HOIL1 does not seem to alter TNF-R signaling.

We have **dedicated a paragraph in the Discussion** (page17/18: 505-529) to describe the implications of our findings that TCR-induced HOIL-1 cleavage influences the balance between TNFR-triggered NF- κ B and cell death signaling.

- The data with primary lymphocytes and TNF α (Fig.1B) are hard to interpret. One could also consider there are no overt effects on I κ B α degradation. Moreover, the effect of TRAF6sg, which serves as a positive control, is rather modest.

In depth analyses of TNF α signaling was not scope of the manuscript, but it served as a control to confirm effective HOIP and HOIL-1 KO. Previous data have shown that HOIP deficiency impairs TNF α -induced NF- κ B signaling (e.g. 25284787, 26670046). In contrast, absence of TRAF6 does not affect TNF α -induced NF- κ B activation in Jurkat T cells (34767456) and other cells (10215628). We show that I κ B α degradation in primary human T cells is partially impaired by HOIP and HOIL-1 deficiency, best detected in single-cell analyses (**Figure 1C, S1F**). In contrast, TRAF6 KO does not affect TNF α -induced I κ B α degradation (**Figure 1C, S1F**). Thus, our data in human primary and Jurkat T cells are in line with the published results.

Reviewer #2 (Remarks to the Author):

Manuscript by Gross and colleagues examines the role of TRAF6 and LUBAC components in Card11-Bcl10-Malt1 (CBM) complex function and signaling. The authors used primary and transformed human cells with knockout of TRAF6 and LUBAC components to explore their importance for CBM induced NF- κ B and MAPK activation and regulation of proteolytic activity of MALT1. The authors conclude that TRAF6 is critical for CBM induced NF- κ B and MAPK signaling while LUBAC play a minor, more supportive role. However, LUBAC is relevant for the modulation of MALT1 proteolytic activity and substrate preference.

Overall, experiments were well designed, technically well executed and conducted with appropriate controls. However, conceptual novelty is minimal as the authors focus on defining and/or confirming details and specifics involving CBM complex signaling such as lysine residues on Bcl10 that are primary targets of ubiquitination, or residues that are important for the proper complex assembly.

Our comparative analysis how TRAF6 and LUBAC affect CBM-dependent T cell signaling and activation provides new key findings how these E3 ligases modulate immune responses. The major advances are summarized in the initial remarks for all reviewers. We now provide **new transcriptomic analyses (new Figure 1G-I, S2)** to support and extend our conclusions. Importantly, we strongly believe that our thorough comparative and mechanistic analysis is timely, and the results are critically revealing how different ubiquitination events controls T cell activation. We are certain that our report will be influential for many future studies.

What is missing is the biological-functional relevance of signaling events studied in this manuscript. When they knockout TRAF6 and LUBAC components the authors biochemically evaluate and quantify signaling readouts (I κ B α phosphorylation and degradation, or p65 phosphorylation, for example), which is a proper way of evaluating signaling pathways. However, we do not know how this is reflected on NF- κ B mediated gene expression and cytokine production. Therefore, in figures where they biochemically evaluate signaling readouts, the authors should also check gene expression and cytokine production stimulated by CBM or TNFR complexes (this applies to figures 1, 2, 6, S6, etc).

We provide more functional data on the impact of TRAF6 and LUBAC and **performed gene expression profiling using RNA sequencing** in human CD4 T cells (**new Figure 1G-I; S2**). We focused and analyzing induction of NF- κ B target genes in human and Jurkat T cells (**new Figure 1H, 1I, 2N, S3I and 6E**).

In primary human T cells, we cannot obtain a complete KO in all CD4 T cells, but we achieve between 40-50% homozygous KO (**Figure S1A, B**). Nevertheless, we see that sgTRAF6 is causing a significant reduction of the NF- κ B target gene signature in primary human CD4 T cells (**Figure 1H**). While HOIP sgRNA does not have a significant impact on the NF- κ B signature in total, induction of distinct NF- κ B target genes is also HOIP dependent (**Figure 1H, I and S2F**). Even though LUBAC-deficiency does not have a significant impact on CBM signaling and NF- κ B p65 translocation in human CD4 T cells (**new Figure 1E, F**), it can affect induction of NF- κ B target genes, and we **modified the Discussion** on page 14: 402-409; page 15: 436-441.

To verify the functional relevance of TRAF6 and HOIP for NF- κ B induction in Jurkat T cell, we **determined expression of classical NF- κ B target genes** NFKBIA/I κ B α and TNFAIP3/A20. TRAF6 and not HOIP is required to activate both genes after T cell activation with CD3/CD28 or PMA/Iono. Vice versa, HOIP and not TRAF6 is critical for upregulation of these genes in response to TNF α (**Figure 2N, S3I**). Further, as expected given the critical role of ubiquitin acceptor lysines in BCL10 for CBM complex assembly, mutation of these sites abolishes NFKBIA/I κ B α and TNFAIP3/A20 induction to the same degree as a known BCL10 oligomerization mutant (**Figure 6E**).

NF- κ B plays a pivotal role in T cell activation, cytokine induction and cell fate decisions. The scope of our study was to decipher the exact mechanisms by which E3 ligases tune upstream signaling to canonical NF- κ B in human T cells. We also included new mechanistic data showing that TRAF6 and HOIP regulate MALT1 substrate selection (**new Figure 3E-J**), which of course is also indirectly affecting NF- κ B activation. We have **revised the Abstract, Results and Discussion** (marked in yellow) to adequately integrate our new findings and discussed the conceptual implications of our work.

The authors should clearly state that they did not perform cryoEM structural experiments, but they merely interpret or model existing structural data.

We never wanted to give the impression that we report here a new BCL10-MALT1 cryo-EM structure, and we apologize for any misunderstandings. To avoid any confusions, we **now changed the wording to 'published' cryo-EM** in the Results (page 13: 348) and we also added the reference to the figure legend 7A, where so far only the PDB number was mentioned. We do not see it as a weakness but rather a strength to re-use published data to address novel crucial research questions.

Reviewer #3 (Remarks to the Author):

In signaling via lymphocyte antigen receptors, two ubiquitin ligases, TRAF6 and LUBAC, plays crucial roles. In T lymphocytes, ligation of CD3 and CD28 (CD3/CD28) or stimulation with PMA and ionomycin (P/I) induces formation of CARD11-BCL1-10-MALT1 (CBM) complex and elicits activation of NF- κ B and MALT1 protease (paracaspase). The authors dissected the involvement of the two ubiquitin

TRAF6 and LUBAC in CBM-mediated signaling in peripheral human T cells and Jurkat T cells. LUBAC is composed of three subunits, HOIP, which is the catalytic subunit for linear ubiquitylation, HOIL-1, and SHARPIN. The catalytic subunit is HOIP and no LUBAC exist in cells lacking HOIP. The amount of HOIP is destabilized in cells lacking HOIL-1 and/or SHARPIN, whereas HOIL-1 or SHARPIN can exist without HOIP. In human cells, although loss of HOIL-1 heavily destabilized HOIP, the small amount of HOIP can exist as HOIP can form complex with SHARPIN. The authors showed that TRAF6 is prerequisite for NF-kappaB activation via the CBM complex, whereas LUBAC is dispensable. The authors have also shown that HOIP is involved in the selection of substrates of the MALT1 protease. Some observations described in this manuscript have been reported previously or some molecular mechanism of signaling have already reported in other signaling pathways in which TRAF6 and/or LUBAC are involved. Involvement of both ligases in substrate selection and activation of MALT1 protease is interesting, however, the molecular mechanism underscoring the substrate selection has not been dissected. Thus, the reviewer feels that the results presented in this manuscript might be informative in the field of immunology, however, the manuscript might not be suitable for a high impact general journal such as Nature Communications. The specific comments are listed below.

We thank the reviewer for the insightful comments. In fact, there have been conflicting data concerning the role of LUBAC and specifically HOIP catalytic activity for CBM complex signaling in Jurkat T cells. While one study suggested that HOIP drives NF- κ B activation independent of its catalytic activity (Dubois 2014, 24497531), another study suggested that E3 ligase activity is required (Yang 2016, 27777308). Also there have been some discrepancies regarding the role of SHARPIN, either being or not being involved (Thys 2021, 33392484; Yang, 2016, 27777308). To further clarify this, we **have mentioned these discrepancies** in the **Introduction** (page 3: 66-73) and in the **Discussion** (page 15: 422-441).

Our aim was to perform comprehensive analyses in primary human CD4 T cells and Jurkat T cells in a clear genetic setting using straight knock-out. We acknowledge that residual HOIP may still drive residual NF- κ B activation in an E3 ligase-dependent or -independent manner in the absence of HOIL-1 or SHARPIN. However, our sgHOIP achieved efficient HOIP targeting in primary human T cells (see also comments below). Thus, effects on signaling should be detectable independent of whether HOIP acts as a scaffold or as an enzyme. The findings are further supported by the complete HOIP KO and reconstitution in Jurkat T cells.

We also provide more mechanistic insights regarding the role of HOIP and TRAF6 for MALT1 substrate selection (see below).

1. It has been already shown that the linear ubiquitylation activity of LUBAC is dispensable for the CBM mediated NF-kappaB activation although TRAF6 is indispensable. The authors showed that using human primary CD4 T cells, LUBAC is dispensable for NF-kappaB activation in Fig 1. They tried to knock out the components of LUBAC including HOIP and HOIL-1, however substantial amounts of them still remained. Since LUBAC is shown to function as a scaffold complex to facilitate NF-kappaB activation in the CBM signaling pathway despite the dispensability of linear ubiquitylation, the results cannot rule out the possibility that LUBAC plays a role in CBM-mediated NF-kappaB signaling as a scaffold. It is worth noting that linear ubiquitylation activity of LUBAC is indispensable for NF-kappaB activation in TNF signaling, which is completely different situation from the CBM signaling. The authors tried to evaluate the NF-kappaB activation using flowcytometric analysis to detect the amount of IkappaBalpha. However, they did not evaluate the amount of HOIP in sgHOIP treated cells. Therefore, they cannot rule out the possibility that HOIP is still expressing in sgHOIP treated cells because of the inefficient knockout.

We performed new analyses to address these questions. Technically, we cannot achieve a complete KO of LUBAC components or TRAF6 in primary human T cells. Thus, main conclusions have been based on single cell CD4 T cell assays, which allows to monitor differences even in a heterogenous cell population. sgTCR α transfection indicated that we are able achieve a homozygous KO efficiency of ~50% (**new Figure S1A**). Using TRAF6 staining by flow cytometry, we show a partial depletion by sgTRAF6 (**new Figure S1B**). Importantly, in line with the approximate KO efficiency, ~40% of the sgTRAF6 transfected CD4 T cells no longer react to P/I stimulation with I κ B α degradation, p65 phosphorylation and p65 translocation (**new Figure 1E, F**), providing strong evidence that T cells with a homozygous depletion of TRAF6 are unable to activate canonical NF- κ B. We have tested several antibodies, but unfortunately, we lack antibodies that detect HOIP or HOIL-1 in flow cytometry. However, based on Western blotting we clearly see that targeting of HOIP and HOIL-1 is very efficient. Especially HOIP KO is even more efficient than TRAF6 KO (**Figure 1A, S1C, S2A**). Given the strong reduction by sgHOIP in Western blots (80-90% reduction of the protein), there must be CD4 T cells with homozygous deletion of HOIP. However, in all single measurements (I κ B α degradation, p65 phosphorylation and p65 translocation), we were unable to detect diminished NF- κ B activation in response to P/I stimulation. The data have been confirmed after complete HOIP KO in Jurkat T cells.

Thus, it is valid to conclude that HOIP and LUBAC do not have a major impact on initial NF- κ B signaling. However, we noted in our **new transcriptomic analyses** that there is a contribution of HOIP to high expression of some NF- κ B target genes (see also comment to rev #1 and #2). How HOIP affects gene induction remains to be determined, but effects may rely on the strength of the stimulation and the modulation of MALT1 substrate cleavage. We now **mention this explicitly in the Discussion** (page 14: 402-409; page 15: 436-441).

2. The authors claimed that loss of HOIP only mildly attenuated NF-kappaB activation in Jurkat cells stimulated with P/I or CD3/CD28 in Figure 2D. However, in HOIP KO#8, NF-kappaB activation by both P/I and CD3/CD28 is significantly attenuated because of loss of LUBAC and attenuation is more pronounced in CD3/CD28 stimulation, more physiological stimulation than in P/I, which reflects the scaffold function of LUBAC as no LUBAC exist in HOIP null cells. The authors showed that loss of HOIL-1 has virtually no effect on NF-kappaB activation by P/I or CD3/CD28. Since in cells lacking HOIL-1, the small amount of HOIP can exist and can form the LUBAC complex with SHARPIN in human cells although the amount of HOIP greatly reduced, which clearly indicated that small amount of LUBAC is enough to activate NF-kappaB in CBM signaling as scaffold. Therefore, the authors' claim that LUBAC is dispensable for CBM-mediated NF-kappaB activation is not supported by the results shown in this manuscript.

Indeed, we see a mild decrease in NF- κ B reporter gene activation in one HOIP KO Jurkat T cell clone. However, given that HOIP reconstitution does not augment NF- κ B reporter gene activation (**Figure 2J, K**) or NFKBIA/I κ B α and TNFAIP3/A20 expression (**new Figure 2N**) P/I or CD3/CD28 stimulation, this is rather clonal variation, since T cell responses can considerably vary depending in individual Jurkat clones. In HOIP KO Jurkat T cells, TNF α -triggered NF- κ B activation is reduced (Figure 2D, 2K, 2N), and apoptosis induction is enhanced (**Figure 4D-F**), demonstrating the functional absence of HOIP.

Of note, if HOIP even in small amounts acts independent of its catalytic activity to mediated CBM-triggered NF- κ B activation, the described linear ubiquitination of CBM components and NEMO would be mere bystander effects. Especially in this case, our structure/function analyses are important, because they show that linear ubiquitination of BCL10 cannot mediate NF- κ B activation. Further, it is intriguing and can serve as a new avenue of research that K63- and M1-ubiquitination can control substrate selection.

Acknowledging that there is an impact of HOIP KO especially on transcriptional activation of NF- κ B targets in primary human T cells, we softened our conclusions. We mention in the Discussion that there is an auxiliary role of HOIP and LUBAC (page 15: 436-441).

3. The observation that loss of HOIP affect selectivity of MALT1 substrates although activation of MALT1 is not affected by loss of HOIP although the authors has not examined the molecular mechanism underlying the selectivity. Among the MALT1 substrates, cleavage of HOIL-1 was heavily impaired in HOIP KO cells. Since LUBAC is recruited to the CBM complex, in which MALT1 exists and is activated, upon stimulation with P/I or CD3/CD28. Thus, it seems likely that HOIL-1 is recruited to the CBM complex and cleaved by MALT1 in the complex. However, in HOIP KO cells, HOIL-1 cannot be recruited to the CBM complex, which may result in the inefficient cleavage of HOIL-1 in HOIP KO cells. Therefore, it is worth examining the molecular mechanism underscoring substrate selection. For example, recruitment of MALT1 substrates are affected by loss of HOIP or TRAF6. Also, it is of interest to know linear ubiquitylation activity of LUBAC is involved in the substrate selection.

We agree with the reviewer that it is important to examine the mechanism of substrate recruitment by HOIP and TRAF6. We have been trying to monitor by IP HOIP- and TRAF6-dependent recruitment of endogenous substrates to MALT1, but we have not been able to reliably detect and quantify these most likely very transient interactions. Importantly, we performed **new experiments on the mechanism(s), how HOIP and TRAF6 control substrate cleavage** by reconstituting KO cells with WT and mutant constructs (**new Figures 3E-J**).

Indeed, HOIL-1 cleavage by MALT1 relies on LUBAC assembly, but not HOIP E3 ligase activity (**Figure 3E**). Thus, HOIP controls HOIL-1 cleavage by mediating the recruitment to the CBM complex. In contrast, impaired CYLD cleavage relies on HOIP E3 ligase activity (**Figure 3F**), showing that HOIP uses different mechanisms to control HOIL-1 and CYLD cleavage. Again, Regnase-1 cleavage is not influenced by HOIP WT or mutants (**Figure 3G**).

In the case of TRAF6, the efficacy of inducible HOIL-1, CYLD and N4BP1 cleavage relies on UBC13 recruitment (C70A mutant) and dimerization (R88A/F118A mutant), which both mutations lead to defective TRAF6 E3 ligase activity (**Figure 3H-J**). In case of HOIL-1, this is highly consistent with the observation that M1-ubiquitination of BCL10 and thus LUBAC recruitment to the CBM complex depends on TRAF6 E3 ligase activity (**Figure 5D**) and apparently conjugation of K63 chains also recruits other substrates to MALT1.

Overall, the new data on the mechanism of MALT1 cleavage add important insights into how HOIP and TRAF6 control MALT1 substrate cleavage. We have **amended the Results** (page 8/9: 218-241) **and Discussion** (page 16: 448-466) accordingly.

The reviewer is also very curious to know why loss of TRAF6 induces activation of MALT1.

We are, of course, also keen to unravel how TRAF6 maintains MALT1 protease in an inactive state in resting T cells and we are working on it. We have shown that constitutive MALT1 protease activation in the absence of TRAF6 activity relies on tonic TCR signaling, as well as the presence of CARD11 and MALT1 binding to BCL10 and TRAF6 (O'Neill, 2021; 34767456). Therefore, we hypothesize that TRAF6 acts by counteracting potentially weak constitutive CBM complex assembly that can lead to unwanted auto-activation of T cells in response to tonic TCR signaling. MALT1, BCL10 and CARD11 are potential candidates for TRAF6 ubiquitination, but currently we lack the methodology to look at these events, especially under homeostatic conditions.

Importantly, our manuscript adds new important insights into the intricate role of K63 and linear ubiquitination for CBM complex assembly, activity, and function. Thus, it represents a significant step towards understanding the complex regulation of TRAF6 (and LUBAC) in the system.

4. The authors claimed that cleavage of HOIL-1 by MALT1 attenuate LUBAC function. However, it has been already shown previously (Fung et al Front Immunol. 12:749794, 2021 for example).

Fung et al showed that expression of uncleavable HOIL-1 K165R enhances IL1-induced NF- κ B activation, but the study was carried out in fibroblasts in the absence of any MALT1 activating stimulus. Thus, the influence of MALT1 is unclear and it is very difficult to interpret the data. As mentioned, others have looked at the impact of HOIL-1 cleavage in different settings, but conflicting results regarding a positive or negative effect of HOIL-1 cleavage have been obtained (26525107, 27006117, 26573773).

We use **stable reconstitution of HOIL-1 KO cells**, which achieves expression at endogenous protein levels in >90% of the cells, which is a very clean genetic setting (**Figure 4**). We are not claiming that HOIL-1 cleavage acts by simply attenuating LUBAC functions. We demonstrate that the N-terminal HOIL-1 cleavage fragment binds and stabilizes HOIP and partially rescues TNFR-induced NF- κ B activation. However, despite lower expression, HOIL-1 N-term prevents TNFR-induced cell death to the same extent as HOIL-1 full length. We thereby describe a new finding on how the TCR can crosstalk to the TNFR and potentially other receptors via MALT1 protease. HOIL-1 cleavage does not inactivate LUBAC function at the TNFR but shifts the balance from pro-inflammatory NF- κ B activation to stronger pro-survival function of the LUBAC. It will be worthwhile to explore the effect of uncleavable HOIL-1 in autoimmune/inflammatory or even infection models (see also our response to rev #1).

5. In HOIP KO cells, TRAF6 should be recruited to the CBM complex and conjugate K63 chains. Thus, it is very curious to know why ubiquitylation could not be detected with P4D1 (pan ubiquitin antibody) in HOIP KO cells in Figure 5A.

Indeed, we have only been able to detect M1-linked ubiquitin chains attached to BCL10 after BCL10-IP. In line, also Yang et al 2016 (27777308) mainly detected M1-linked ubiquitin chains. Thus, it seems that M1-linked chains are the main modification conjugated to BCL10. However, since LUBAC can only attach M1-ubiquitin chains to an initiator ubiquitin (mono or chain), potentially short K63-ubiquitin chains may be less stable. Alternatively, the amount of ubiquitin in these short chains may be below the detection limit of the pan-ubiquitin antibody.

6. What is delta CD2, at the bottom of page 7?

The lentiviral constructs contain human Δ CD2 as a surface marker for flow cytometry-based detection of transduced cells. It allows to determine the efficacy and homogeneity of the transduction, and we use only cells with a transduction efficacy of >90%. Δ CD2 does not contain a cytoplasmic tail, so it is devoid of signaling domains. We included the reference in which we have first described the construct (Hadian, 2011, 21622571).

7. What is MALT1 TBM at the bottom of page 9?

MALT1 TBM refers to TRAF6 binding mutant E316A/E806A of MALT1. We changed it to E316A/E806A for clarity.

Reviewer response letter

Reviewer #1 (Remarks to the Author):

The authors satisfactorily addressed my concerns.

We thank the reviewer, and we are happy that we have been able to address all concerns.

Reviewer #2 (Remarks to the Author):

The authors have largely addressed reviewers' comments. I would include the data with SPATA2 KO cells in the supplemental.

We thank the reviewer, and we are pleased about the positive evaluation. As suggested, we now added the data on CYLD cleavage in SPATA2 KO in **Fig. S4g**. We write on page 9 (lines 234-237):

SPATA2 bridges CYLD to LUBAC³⁸, but SPATA2 deficiency did not significantly affect I κ B α degradation or CYLD cleavage in Jurkat T cells, suggesting that SPATA2 is not involved in the recognition of CYLD by the CBM complex and MALT1 (Fig. S4g).

Reviewer #3 (Remarks to the Author):

The authors have made commendable efforts to address the concerns raised in the initial review. Nevertheless, several critical issues remain unresolved, which merit further clarification and investigation.

We thank the reviewer for acknowledging our efforts and are happy that we managed to address many concerns. We address all remaining concerns below in the detailed point-by-point response. Importantly, we added data on the role of A20 ZF4 and ZF7 for MALT1 cleavage. We are confident that we have been able to answer the open questions of the reviewer.

1. While the authors attempted to elucidate the impact of HOIP knockout in human CD4⁺ T cells, they did not provide quantitative data on HOIP expression in these cells (Figure 1B).

We added data showing expression of *TRAF6*, *RNF31/HOIP* and *RBCK1/HOIL-1* in human CD4 T cell subsets (**new Fig. S1a**). The data are taken from the Monaco dataset in Protein Atlas (<https://www.proteinatlas.org>) in which RNA sequencing (RNAseq) of 29 immune cell types from PBMCs of healthy donors was performed (PMID 30726743). All three genes are expressed in the major CD4 T cell subsets found in PBMCs of healthy donors. Of note, *HOIL-1* is most abundant, and *HOIP* shows the lowest expression, comparing normalized transcripts per million (nTBM). Protein expression is detected for all three genes (see **Fig. 1b**), showing that TRAF6 and LUBAC subunits are expressed in human CD4 T cells.

Moreover, we determined the efficacy of TRAF6 and HOIP depletion by quantifying protein amounts based on Western blotting after sgRNA transfection of CD4 T cells from different donors (n=7). On average, ~60% reduction of TRAF6 protein and ~85% reduction of HOIP protein was achieved with the respective sgRNAs (**new Fig. 1a**).

We adjusted the first paragraph on page 5 (line 99 onward) to add these results.

Furthermore, although they assert that HOIP deficiency attenuates TNF- α -mediated NF- κ B activation, the observed reduction in I κ B α levels upon TNF- α stimulation (Figure 1C) appears inconsistent with this claim and warrants further explanation.

Indeed, we find that HOIP or HOIL-1 KO attenuates but does not abrogate TNF α -induced NF- κ B signaling in human CD4 T cells (**Fig. 1d, S1e, S1g**). Previous reports have shown that absence of HOIP in human and murine fibroblasts reduces but does not abolish NF- κ B signaling and activation (e.g. PMIDs 25284787 (Peltzer et al, 2014) and 26008899 (Boison et al, 2015)). It was not our main focus to study the contribution of LUBAC for TNFR signaling in primary T cells, but we think the data are quite consistent with what has been published in other systems. To further explain this, we revised the discussion and write on page 15 (lines 428-431):

Consistent with previous data on LUBAC deficiency in human and murine fibroblasts^{33,45}, TNFR-induced NF- κ B activation was reduced but not abolished by HOIP or HOIL-1 ablation in human T cells, suggesting impaired LUBAC functions.

2. The authors suggest that HOIP may be dispensable for TCR-mediated NF- κ B activation in primary T cells. However, this notion has already been reported in the literature (Okamura et al., Scientific Reports, 2016), and the current manuscript does not offer substantial novel insights beyond this established finding.

Okamura et al generated a conditional mouse model in which a HOIP ^{Δ linear} fragment lacking the catalytic domain is expressed under control of CD4-Cre. They write in the abstract: 'HOIP ^{Δ linear} CD4+ T cells failed to phosphorylate I κ B α and JNK through T cell receptor-mediated stimulation.' This is the opposite of what we report here.

There are several relevant points: (1) They did not analyze a HOIP KO but a truncation mutant, which may have very different, possibly even dominant negative effects. (2) Their conclusions about TCR-induced NF- κ B signaling are based on representative experiments using T cells from one HOIP ^{Δ linear} and one control mouse. No empirical data from independent experiments have been provided. Thus, it is not possible to draw clear conclusions from these data. (3) CD4 and CD8 T cell numbers are severely reduced in HOIP ^{Δ linear} mice and T cells undergo apoptosis. Thus, it is well possible that the effects on T cell signaling may be linked to impaired T cell viability.

Here, we have taken a different approach by performing *ex vivo* KO of TRAF6 and HOIP KO in human CD4 T cells. The CD4 T cells are viable after KO and by this we avoid the issues of defective differentiation and impaired viability of T cells, which has been associated with conditional LUBAC deficiencies in mice (PMIDs 27857075 (The et al, 2016) and 27786304 (Okamura et al, 2016)). We mention and discuss both publications and their findings in the context of our data. Importantly, by showing that LUBAC is largely dispensable for CBM signaling in human CD4 T cells, we come to different conclusions about the role of LUBAC in T cells. Thus, the comparison to Okamura et al shows that we are providing novel insights beyond established findings.

3. The authors have appropriately addressed the substrate specificity of MALT1. The finding that HOIP is required for MALT1 cleavage is intriguing. However, the molecular mechanism underlying this requirement remains unexplored. It is conceivable that A20 is recruited to MALT1 via recognition of M1-linked ubiquitin chains conjugated to the CBM complex, potentially mediated by the ZF7 domain of A20, which is known to bind M1-linked chains. This hypothesis deserves experimental validation.

We have explored the molecular mechanism by reconstituting HOIP KO Jurkat T cells with HOIP mutants either defective in LUBAC assembly (Δ UBA) or catalytic activity (C885S). We show that LUBAC assembly is necessary and sufficient for facilitating HOIL-1 cleavage, while HOIP E3 ligase activity is necessary to impair CYLD cleavage (**Fig. 3e, f**). Very similar

experiments have been conducted to address the molecular mechanism underlying the requirement of TRAF6 for cleavage of distinct MALT1 substrates (**Figure 3h-j**).

As suggested, we performed additional experiments to explore mechanistically how ubiquitination may control A20 cleavage by MALT1. As noted by the reviewer, zinc finger 7 (ZnF7) can bind M1 ubiquitin chains (generated by HOIP) and zinc finger 4 (ZnF4) can bind K63 chains (generated by TRAF6). We reconstituted A20 KO Jurkat T cell with A20 WT, ZnF4 mut, ZnF7 mut and ZnF4/7 double mut (**new Fig. S4h**). We quantified cleavage of A20 as well as of the substrates CYLD and HOIL-1 after P/I stimulation (**new Fig. S4i**). Indeed, cleavage of A20 was severely impaired in ZnF4, ZnF7 or ZnF4/7 mutant Jurkat T cells. Thus, the conjugation of M1 (HOIP) and K63 (TRAF6) ubiquitin chains is necessary for efficient A20 cleavage by MALT1. Accordingly, we have shown that the A20 ZnF4/7 mutant is impeded in binding to the CBM complex (PMID 35099607 (Yin et al, 2022)). MALT1 cleavage of HOIL-1 or CYLD cleavage was not affected by ZnF mutations, excluding that A20 functions in a more general term as a bridging factor of ubiquitination-dependent substrate recognition. As in previous experiments, we provide empirical data from multiple replicates on the effects of A20 mutations on substrate cleavage. We modified the results on page 9 (lines 237-244):

*A20 zinc finger (ZnF) 4 and 7 bind to K63- and M1-linked ubiquitin chains, respectively³⁹. Thus, we asked if A20 recruitment to ubiquitin chains via the ZnFs controls cleavage of A20 or even HOIL-1 and CYLD by MALT1. We reconstituted A20 KO Jurkat T cells reconstituted with A20 WT and ZnF mutants (**Fig. S4h**). Cleavage of A20 was severely impaired in ZnF4, ZnF7 or ZnF4/7 mutant expressing Jurkat T cells, suggesting that TRAF6 and HOIP conjugation of K63- and M1-chains, respectively, controls A20 recognition by MALT1 (**Fig. S4i**). HOIP- and TRAF6-dependent cleavage of HOIL-1 or CYLD cleavage by MALT1 was not affected by A20 ZnF mutations.*

and the discussion on page 16 (lanes 469-474)

Further, proteasome- and MALT1-dependent A20 proteolysis in Jurkat T cells is impeded in the absence of TRAF6 or HOIP, as well as upon mutation of A20 ZnF4 and ZnF7, which bind K63- or M1-ubiquitin chains, respectively. Thus, ubiquitin conjugation and binding are critical for A20 cleavage, suggesting that both E3 ligases fine tune A20 protein expression at the post-translational level after T cell activation³⁷.

to incorporate these interesting findings.

To our knowledge, we show for the first time that K63- or M1-linked ubiquitination is involved in MALT1 substrate selection. Of course, we raise new questions about how ubiquitination controls MALT1 substrate recognition and cleavage: What targets are ubiquitinated to control MALT1 substrate selection? Are other E3 ligases besides TRAF6 and HOIP involved in directing the cleavage of MALT1 substrates? May ubiquitination serve as a mechanism for MALT1 substrate selection beyond the cleavage site and may this knowledge help us to predict new MALT1 substrates? We think that our findings will inspire future research on MALT1 protease.

4. The authors claim that ubiquitin acceptor lysines in BCL10 are essential for CBM complex assembly and function, based on mutational analyses. However, the possibility that these lysine mutations may alter the structural conformation of BCL10, thereby affecting its function independently of ubiquitination, cannot be excluded and should be considered.

This is indeed a very important finding of our study. We provide detailed structure-function analyses of the BCL10-BCL10 and BCL10-MALT1 interphases focusing on the positioning and mutation of lysine residues potentially modified by linear ubiquitination. We demonstrate that ubiquitination at BCL10 K17, K31 and K63 is mutually exclusive with BCL10-MALT1

binding or BCL10 oligomerization and that even conservative K/R mutagenesis affects CBM complex formation. On the other hand, BCL10 ubiquitination relies on CARD11, revealing that CBM complex formation is a prerequisite for ubiquitination. We would like to refer to the discussion, in which we have dedicated an entire paragraph on page 17 (lines 480-517) to these finding and their implications. We speculate based on the BCL10-MALT1 filament structure that ubiquitination at these lysine residues may restrict the growth and length of filaments in cells. However, we also mention the limitations of our study by saying explicitly:

'Further, if the same residues in BCL10 that are ubiquitinated are controlling BCL10-MALT1 filament assembly, a limitation of the study is that current mutagenesis methods do not allow us to prove that these lysines are the direct ubiquitin acceptor sites. Mass spectrometry approaches failed to identify ubiquitination at these residues, but this does not exclude that they can be ubiquitinated in activated lymphocytes²⁷.' (lines 511-515).

Thus, we are very aware of this, and we consider it an important outcome of our study, particularly in guiding future research on the cellular regulation of the CBM complex.

Point-by-point response

REVIEWERS' COMMENTS

Reviewer #3 (Remarks to the Author):

The authors satisfactorily addressed the concerns.

We are happy about the positive evaluation of all reviewers.